# Genetic gradual reduction of OGT activity unveils the essential role of O-GlcNAc in the mouse embryo

Sara Formichetti[1,2], Agnieszka Sadowska[1], Michela Ascolani[1], Julia Hansen[1], Kerstin Ganter[1], Christophe Lancrin[1], Neil Humphreys[1], Mathieu Boulard[1]*

1 Epigenetics & Neurobiology Unit, EMBL Rome, European Molecular Biology Laboratory, Italy,
2 Collaboration for joint PhD degree between EMBL and Heidelberg University, Germany

* matthieu.boulard@embl.it

**Data Availability Statement:** All raw RNA-Seq data generated in this study are available in Biostudies under accession codes E-MTAB-13298, E-MTAB-13297, E-MTAB-13299, E-MTAB-13499. Scripts

## Abstract

The reversible glycosylation of nuclear and cytoplasmic proteins (O-GlcNAcylation) is catalyzed by a single enzyme, namely O-GlcNAc transferase (OGT). The mammalian *Ogt* gene is X-linked, and it is essential for embryonic development and for the viability of proliferating cells. We perturbed OGT's function *in vivo* by creating a murine allelic series of four single amino acid substitutions, reducing OGT's catalytic activity to a range of degrees. The severity of the embryonic lethality was proportional to the extent of impairment of OGT's catalysis, demonstrating that the O-GlcNAc modification itself is required for early development. We identified hypomorphic *Ogt* alleles that perturb O-GlcNAc homeostasis while being compatible with embryogenesis. The analysis of the transcriptomes of the mutant embryos at different developmental stages suggested a sexually-dimorphic developmental delay caused by the decrease in O-GlcNAc. Furthermore, a mild reduction of OGT's enzymatic activity was sufficient to loosen the silencing of endogenous retroviruses *in vivo*.

## Author summary

Posttranslational modifications of proteins enable cells to rapidly adapt to changes in their environment. Here, we investigated the role of the modification of intracellular proteins by a sugar molecule in the mouse embryo. The O-GlcNAc transferase (OGT) adds the sugar O-GlcNAc to thousands of nuclear and cytoplasmic proteins, and it is essential for the development of the mammalian early embryo. To interrogate its biological functions *in vivo*, we created several mouse models bearing precise *Ogt* mutations that cause a reduction of OGT activity to a previously determined range of degrees, without interfering with OGT protein expression. We found that the severity of the developmental phenotype scales with the degree of impairment of OGT activity, demonstrating that the O-GlcNAc modification itself, and not only the presence of OGT, is required for mouse embryonic development. We show how different levels of cellular O-GlcNAcylation are critical for different developmental stages, and that even a mild reduction perturbs the silencing of retrotransposons. The dosage of OGT can vary between sexes because the *Ogt*

and custom Snakemake pipelines are available at
https://github.com/boulardlab/Ogt_mouse_models_Formichetti2024.

**Funding:** This research was supported by the European Molecular Biology Laboratory (EMBL) to MB. The funders had no role in study design, data collection and analysis, decision to publish, or preparation of the manuscript.

**Competing interests:** The authors have declared that no competing interests exist.

gene is located on the X chromosome and escapes X inactivation in female extraembryonic tissues. Accordingly, we found that the male blastocyst and placenta are more sensitive to *Ogt*'s impairment.

## Introduction

Over five thousand of nuclear and cytoplasmic proteins are reversibly modified at specific serine and threonine hydroxyls by the covalent linkage of the monosaccharide O-linked-β-N-acetylglucosamine (O-GlcNAc) [1]. While it remains unclear how O-GlcNAc modulates the function of its target proteins [2,3], this posttranslational modification has been implicated in various cellular functions, including transcription [4,5], the cell cycle [6], translation [7,8] and glycolysis [9,10]. O-GlcNAc homeostasis is controlled by a single pair of highly conserved enzymes: O-GlcNAc transferase (OGT) [11,12] and O-GlcNAc hydrolase (OGA) [13], both encoded by a single gene in the majority of animals [14]. The mammalian *Ogt* gene is X-linked, and it is essential for early development [15]. Furthermore, the maternal inheritance of an *Ogt*-null allele results in preimplantation lethality, even in the presence of a functional paternal *Ogt* gene in heterozygous female embryos [16]. Hence, OGT or O-GlcNAc (or both) are necessary for oocyte maturation or for the cleavage stage embryo. Another obstacle to genetic studies in the mouse is the essential nature of *Ogt* for proliferating cells [17], implying that conditional knock-out alleles eventually lead to cellular lethality that can confound molecular phenotypes. The outstanding resistance of *Ogt* to classic genetics approaches has hindered the understanding of the biology of O-GlcNAc during mammalian development, leaving salient questions unanswered. Firstly, it is not demonstrated whether the essential function of *Ogt* in the early embryo resides in its catalytic activity. In a cellular model, the separation of OGT's functions showed that non-catalytic functions also contribute to the cellular phenotype resulting from OGT's absence [18]. Secondly, the molecular roles of OGT and O-GlcNAc in the early mammalian embryo *in vivo* have barely been explored.

A growing body of evidence points towards O-GlcNAc acting as a mediator connecting external cues to the control of development and growth, both at the cellular and organismal level [17,19]. O-GlcNAc's donor substrate, UDP-GlcNAc [20], is the end product of the metabolic hexosamine biosynthetic pathway, which utilizes 2–3% of intracellular glucose [21] and is sensitive to nutrient levels [21,22]. In adult mice, O-GlcNAc is involved in regulating energy conservation [23] and eating behavior [24]. In a murine model of diabetic pregnancy, it partly mediates the negative effect of hyperglycemia [25]. *Ogt* is a facultative escapee of X chromosome inactivation (XCI) that variably escapes random XCI *in vitro* in differentiating cells [26,27], and imprinted XCI (iXCI) during embryonic development [28–30]. iXCI refers to the selective inactivation of the paternally inherited X chromosome that starts in the mouse preimplantation embryo at the 4-cell stage and from the blastocyst onward is maintained only in the extraembryonic tissues [31,32]. Notably, *Ogt*'s escape from XCI is particularly prominent in the postimplantation extraembryonic ectoderm (ExE), the extraembryonic tissue precursor to the placenta, which is the embryonic interface to the maternal environment, providing the embryo with gas, nutrients and hormones. The resulting double dose of OGT in the female's placenta explains the higher resistance of female mouse embryos to maternal stress compared to males [33,34]. In humans, single amino acid substitutions of *OGT* are associated with X-linked intellectual disability (XLID; OMIM #300997), a neurodevelopmental syndrome with heterogeneous symptoms that includes decreased intellectual ability (IQ<<70), low birth weight, short stature, drooling, compromised language skills and often anatomical brain and

body anomalies [35]. OGT-XLID mostly affects males, as random XCI mitigates the effects of the mutations in females by leading to mosaicism of wild type and mutated alleles [35].

O-GlcNAc modifies numerous DNA-associated proteins, counting RNA polymerase II [36], chromatin modifiers [37] and various transcription factors, including the pluripotency master regulators OCT4 and SOX2 [5]. OGT stably binds and glycosylates the Ten-eleven translocation (TET) enzyme which converts 5-methylcytosine (5mC) into 5-hydroxymethylcytosine (5hmC), the first step of 5mC re-conversion to unmodified cytosine (C) [38–41]. DNA methylation is particularly dynamic during preimplantation development, and plays an essential role in the repression of parasitic DNA sequences known as retrotransposons [42]. Recent evidence implicates OGT and local O-GlcNAcylation of chromatin factors in the stable retrotransposon silencing: *Ogt* conditional deletion in mouse embryonic stem cells (mESCs) causes a low-magnitude upregulation of many retrotransposon families [43]. Targeted de-GlcNAcylation at the promoters of the most active murine endogenous retroviruses, namely intracisternal A-particle elements (IAPs), led to their full-blown reactivation [37]. It is therefore tempting to speculate a causal relationship between O-GlcNAc and developmental epigenetic silencing, a hypothesis which remains to be tested *in vivo*. In the fly, *Ogt* is necessary to silence homeotic genes during the larval stages [4,44], through the O-GlcNAcylation of the repressor Polyhomeotic [45]. Whether this functional connection between O-GlcNAc and developmental gene expression is conserved in the mammalian embryo has also not yet been thoroughly investigated.

All the above urge the creation of alternative O-GlcNAc perturbation approaches in the mouse embryo that overcome the cellular and embryonic lethality resulting from *Ogt*'s disruption. To address this challenge, we created a series of mouse models with OGT's catalytic activity impaired to a range of degrees previously determined *in vitro*, by substituting key amino acids in the catalytic core. By analyzing the inheritance of the *Ogt*-hypomorphic alleles, we discovered that OGT's catalysis, hence the O-GlcNAc modification itself, is required for preimplantation development. We then profiled the transcriptional changes resulting from reduced O-GlcNAcylation in two of these hypomorphic mutants. In the preimplantation embryo, we provide the first *in vivo* evidence that a mild decrease in O-GlcNAc causes the transcriptional upregulation of retrotransposons. In the postimplantation embryo, we found a sexually-dimorphic defect in placental differentiation upon lower O-GlcNAc levels. In addition, we created a mouse model bearing an *Ogt*-degron system for the induced fast removal of the OGT protein. Although the system was inefficient in preimplantation embryos grown *ex vivo*, it was effective in primary mouse embryonic fibroblasts (MEFs), where we characterized the immediate transcriptional changes and their evolution over time after acute OGT depletion.

## Results

### Gradual reduction of OGT's catalytic activity proportionally impacts embryonic survival

We leveraged structural and biochemical knowledge on OGT [46] and used CRISPR-Cas9 genome editing in the mouse zygote to create a series of *Ogt* hypomorphic mutants bearing single amino acid substitutions in the core of the catalytic domain: specifically, H568A located in the N-Cat domain and Y851A, T931A and Q849A located in the C-Cat domain (Fig 1A and 1B). These mutations hinder to various extents OGT's binding to its donor substrate UDP-GlcNAc or the mechanism of catalysis [46]; they are unlikely to affect the substrate recognition because substrate specificity is operated by two other domains, namely the intervening domain [47] and the tetratricopeptide repeat (TPR) [48]. The point mutations selected—Y851A, T931A, Q849N, H568A —were previously shown *in vitro* to reduce OGT's enzymatic

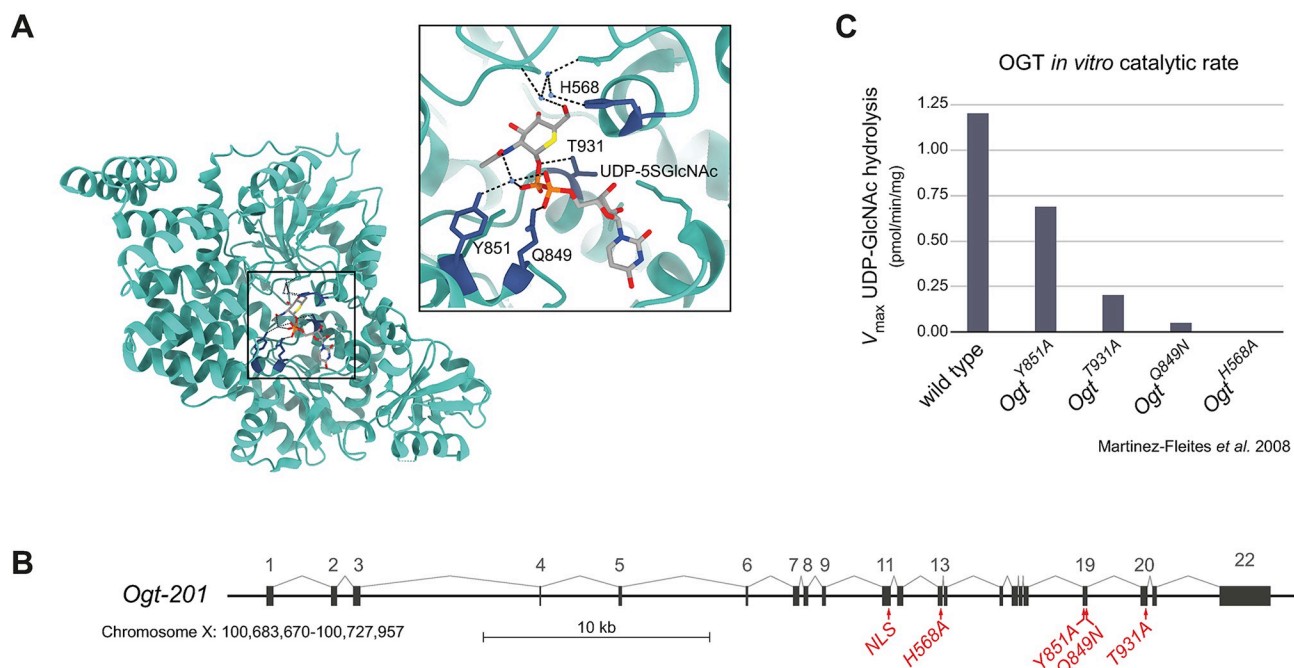

**Fig 1. Structure-guided generation of *Ogt*'s hypomorphic allelic series.** (**A**) Crystal structure of human OGT isoform 1 (UniProt O15294-1) in complex with UDP-5SGlcNAc (PDB 4GYY). The area in the square is shown as an insert with higher details. Except for two single amino acid substitutions outside the catalytic core, human OGT1 is identical to *Mus musculus* OGT isoform 1 (UniProt Q8CGY8-1), hence residues' numbering is the same. The residues mutated in this study are shown in sticks representation in blue and UDP-5SGlcNAc in sticks representation, coloured by heteroatom. T931 and Q849 establish direct interactions with the donor substrate. Y851 interacts with the donor substrate via hydrogen bonds with an intermediate water molecule. H568 is considered to be the catalytic base and in the crystal structure it coordinates different water molecules near the binding site. (**B**) Scheme of the murine *Ogt* genomic context with the location of the introduced point mutations. The exons of the longest transcript *Ogt-201* are numbered. (**C**) Catalytic rate measured *in vitro* for the four studied OGT single amino acid substitutions. Data are from Martinez-Fleites *et al.* 2008 [46] and for H568A also from Lazarus *et al.* 2011 [111]. (**D**) For the four hypomorphic mutants of OGT described in (**A-C**): number of animals ($F_0$) bearing the correct point mutation after CRISPR-targeted mutagenesis in the zygote (founders); number of mutants ($F_1$) produced by female founders (female germline transmission, i.e. maternal transmission); lethal phenotypes observed from the $F_1$ generation onwards.

rate ($V_{MAX}$) to respectively 70%, 20%, 5% and 0% (Fig 1C), and OGT's enzymatic activity ($V_{MAX}/K_M$) to 24%, 18%, 2%, 0% of that of the wild type (WT) protein [46]. To our knowledge, there is no report on the impact of these mutations on cellular O-GlcNAc levels, and thus the developmental phenotypes resulting from their murine alleles were unpredictable.

We observed a clear correlation between the degree of impairment of OGT's activity and the severity of the developmental phenotype, as measured by the success of gene targeting in

the zygote and the rate of germline transmission of the mutant allele (Fig 1D, first two columns). Specifically, no CRISPR-edited founders were obtained at weaning for the mutation of the catalytic base (H568A), indicating that the complete disruption of the glycosyltransferase activity is incompatible with embryonic development for both hemizygous and heterozygous mutants. None of the four founders (3 males and 1 female) obtained for the second most disruptive mutation (Q849N) transmitted the mutation to their progeny, indicating that 2% of OGT's activity is not compatible with early embryonic development and/or for function of germ cells.

For $Ogt^{T931A}$ (retaining ~20% of catalytic rate and activity), the female $F_0$ founders transmitted the mutation to the $F_1$ generation, albeit at a much lower rate than expected, and only heterozygous females were recovered at weaning (postnatal day 21; Figs 1D, third column, and 2A; chi-squared test's p-value = 1.94 $e^{-06}$). This phenotype prompted us to study $Ogt^{T931A}$ mutants at the blastocyst stage.

In contrast, the least disruptive mutation, namely *Y851A*, was maternally transmitted to the $F_1$ generation at normal Mendelian ratio. Of note, *in vitro* OGT(Y851A) retains only ~24% of WT $V_{MAX}/K_M$ (similar to the embryonic lethal T931A), but ~70% of WT $V_{MAX}$. The severity of embryonic lethality being proportional to the decrease in $V_{MAX}$ rather than in the binding affinity suggests that *in vivo* the availability of the donor substrate is not the limiting parameter. Hemizygous $Ogt^{Y851A/Y}$ and homozygous $Ogt^{Y851A/Y851A}$ animals were seemingly healthy, but displayed sub-lethality at weaning (Figs 1D, third column, and 3A). We used this allele to address the function of *Ogt* post implantation, focusing on the placenta.

In addition, we assessed the previously reported OGT's nuclear localization signal (NLS) consisting of the tripeptide $DFP_{461-463}$ [49], by creating a murine allele where $DFP_{461-463}$ is substituted by AAA ($Ogt^{NLS-}$; S1A Fig). This allele aimed at separating the nuclear and cytoplasmic functions of OGT. However, the mutation of this putative NLS had no measurable effect on OGT nuclear localization on primary MEFs (S1B and S1C Fig). Since we found no evidence that the DFP motif is necessary for OGT nuclear translocation, the $Ogt^{NLS-}$ allele was not suitable to address the nuclear function of OGT.

Based on the sub-Mendelian inheritance of the catalytically hypomorphic alleles and on the severity of their developmental phenotype being proportional to the degree of reduction of OGT's enzymatic activity, we conclude that the O-GlcNAc transferase's function of OGT, thus the O-GlcNAc modification itself, is essential for embryonic development, independently of the presence of the OGT protein.

## Maternal inheritance of a severe hypomorphic *Ogt* mutation causes preimplantation sub-lethality

*In vitro*, the T931A substitution causes an ~80% reduction in OGT's activity. *In vivo*, this mutation resulted in a significantly sub-Mendelian distribution of the maternally inherited *T931A* mutated allele already at the blastocyst stage (Fig 2A; chi-square test's p-value = 5.13 $e^{-05}$), where only 4 heterozygous females and 2 hemizygous males were found among 39 blastocysts from two $F_0$ $Ogt^{T931A/+}$ females. Because of *Ogt* escaping imprinted XCI, female heterozygous blastocysts can in principle compensate for a hypofunctional maternal copy. Therefore, the observed preimplantation lethality of $Ogt^{T931A/+}$ heterozygous females implies that the maternal inheritance of a catalytically active OGT is required for preimplantation development. Hence, physiological O-GlcNAc levels are essential for oocyte maturation or the oocyte payload of OGT and O-GlcNAc's modified proteins is essential for the cleavage stage embryo before the onset of expression of a catalytically functional paternal OGT, which starts around the 8-cell stage [50].

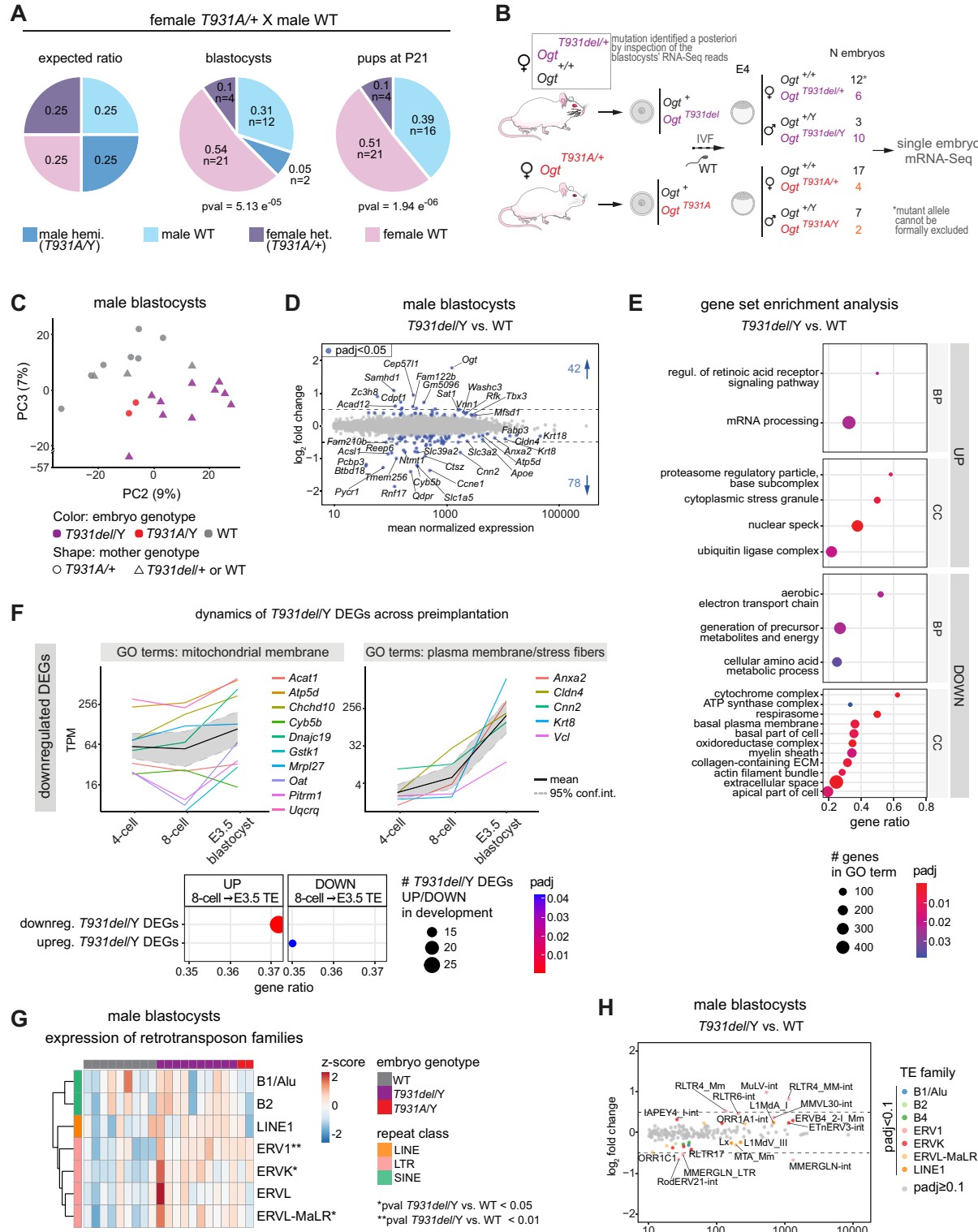

**Fig 2. Maternal inheritance of a severe *Ogt* mutation causes preimplantation sub-lethality and perturbs retrotransposons silencing. (A)** Number and frequency of genotypes for the progeny produced by the mating of female heterozygous $Ogt^{T931A/+}$ with WT males. The left pie chart shows the theoretical Mendelian ratios for an X-linked mutation, the middle and right pie charts show the observed distribution of genotypes at the blastocyst stage and at weaning, respectively. P-values were computed using the chi-squared test for independence. **(B)** Experimental design for the study of blastocysts with maternal inheritance of a mutation at *Ogt*-T931. Oocytes from females $Ogt^{T931A/+}$ and

seemingly $Ogt^{+/+}$ female littermates were fertilized *in vitro* with WT sperm and the embryos grown *ex vivo* to the blastocyst stage (E4), when they were collected for single embryo mRNA-Seq. For the embryos produced by $Ogt^{T931A/+}$ females, the sex and presence of the T931A mutation was determined by PCR genotyping of the cDNA and Sanger sequencing. For the rest of the embryos, the *T931del* allele was identified *a posteriori* by manual inspection of the RNA-Seq reads, and the embryos were genotyped based on sequencing reads mapping to *Ogt*. The WT genotype was assigned based on the absence of reads bearing a mutation at T931. The total number of embryos analyzed per genotype after the *in silico* filtering steps (S5 Table) is indicated. Note that the absence of *T931del*-containing reads cannot formally exclude the presence of this allele within the group of heterozygous females. (**C**) Transcriptomes of individual male blastocysts produced by the experiment in (**B**) in the space defined by PC2 and PC3 of their principal component analysis (PCA), which result in a better separation based on embryonic genotype than PC1 and PC2 (shown in S2B Fig). The variance explained by each PC is shown in parentheses. (**D**) MA-plot from DESeq2 differential expression analysis of single copy genes in $Ogt^{T931del/Y}$ versus WT male blastocysts from all maternal genotypes (N = 10 embryos per embryo genotype). All genes with mean DESeq2-normalized gene counts > 10, adj. p-value < 0.05, any $\log_2$FC (DEGs) are colored, and their number is indicated. Dashed lines show $\log_2$FC = ±0.5. Genes standing out (and with abs($\log_2$FC) ≥ 0.2) are labeled. (**E**) Gene set enrichment analysis (GSEA) of gene expression change in $Ogt^{T931del/Y}$ versus WT male blastocysts. Among significant Biological Process (BP) and Cellular Component (CC) gene ontology (GO) terms, the first 20 based on Normalized Enrichment Score (NES) are shown. Terms are ordered based on gene ratio. The size of dots is proportional to the number of total genes of a GO term. Gene ratio = fraction of total genes of the GO term which are concordantly changing in mutant embryos. (**F**) Top: expression dynamics of $Ogt^{T931del/Y}$ DEGs belonging to the indicated GO terms throughout preimplantation development. The two biological replicates per stage were averaged. For each gene, the TPM value in the E3.5 blastocyst is the highest TPM value between the values in E3.5 ICM and E3.5 trophectoderm. The mean among all genes is drawn, as well as the 95% confidence interval, computed using basic nonparametric bootstrap (R function 'mean.cl.boot'). Y-axis ticks are in $\log_2$ scale. TPM: Transcript Per Million. Bottom: enrichment analysis of up- and downregulated $Ogt^{T931del/Y}$ DEGs in the genes changing in unperturbed embryos between the 8-cell stage and the trophectoderm of the E3.5 blastocyst (ranked by -log10(adj. p-value) *sign(log2FC) from E3.5TE-vs-8-cell comparison). Gene ratio = fraction of total up-/downregulated DEGs which are up-downregulated between the two preimplantation stages. Top and bottom: mRNA-Seq data for unperturbed embryos are from GSE66582 and GSE76505. (**G**) Heatmap of the expression (in Fragments Per Kilobase Per Million, FPKM) of the main families of retrotransposons in the single male blastocysts produced in (**B**). Values are scaled by rows. Retrotransposon families are clustered based on Euclidean distance and coloured by class. P-values were computed using unpaired Wilcoxon rank sum exact test of FPKM values, and they are indicated when < 0.05. (**H**) MA-plot from DESeq2 differential expression analysis of retrotransposons in $Ogt^{T931del/Y}$ versus WT male blastocysts (N = 9 WT, 10 $Ogt^{T931del/Y}$ embryos). All repeats with mean DESeq2-normalized gene counts > 10, adj. p-value < 0.1, any $\log_2$FC are colored by their class. Repeats standing out are labeled.

The scarcity of blastocysts recovered bearing the *T931A* mutation prevented the statistical analysis of their molecular phenotype. However, because preimplantation development heavily relies on the maternal payload of RNAs and proteins, we assessed the influence of the maternal genotype on the embryonic transcriptome. We performed single embryo mRNA-Seq [51] on blastocysts generated through *in vitro* fertilization (IVF) of oocytes from a pair of $F_0$ $Ogt^{T931A/+}$ females and from a pair of female littermates for which the *Ogt*-T931A mutation had not been detected by PCR genotyping. Both groups were fertilized with the same WT sperm. We manually inspected the RNA-Seq reads mapping to *Ogt* and unexpectedly uncovered the presence of an additional mutation of the T931 codon in the blastocyst population produced by the control females, namely a precise deletion of T931 (S2A Fig). The *T931del* allele was recovered in the blastocyst population with a higher frequency than T931A (Fig 2B). The data suggest that the deletion of T931 is better structurally tolerated than its substitution by alanine, likely because of the next amino acid being also a threonine (T932).

This serendipity gave us the possibility to get access to the transcriptome of blastocysts bearing the deletion of an essential catalytic amino acid [46]. The principal component analysis revealed that the *T931del* mutation impacts the transcriptomes of hemizygous males, but has little effect on that of heterozygous females (on PC3; Figs 2C, S2B and S2C). This sexual dimorphism is likely explained by the expression of the paternally inherited *Ogt* allele (WT) in the blastocyst [29,50], as confirmed by higher levels of *Ogt* transcripts in WT female blastocysts compared to WT males (S2D Fig). In good agreement, male hemizygous mutants show a more dramatic compensatory increase in *Ogt* expression than heterozygous mutant females (S2D Fig). The upregulation of *Ogt* in response to decreased O-GlcNAc levels has been previously reported in cells and tissues [52–57].

Thus, we focused our transcriptional analysis on the males and found 120 differentially expressed genes (adj. p-value < 0.05, any $\log_2$FC, henceforth DEGs) in the $Ogt^{T931del/Y}$

blastocysts when compared to the WT, with 2/3 of the genes downregulated and 90% of the significant changes below 1 $\log_2$FC (Fig 2D). Gene set enrichment analysis revealed the upregulation of proteasomal activity and stress granules along with the downregulation of amino acid metabolism, mitochondrial respiration and both transport and cell adhesion functions associated with the plasma membrane (Fig 2E). We recently found that depleting nuclear O-GlcNAc from preimplantation nuclei slows down development, and that the associated reduction of transcripts involved in aerobic respiration and membrane transport in the blastocysts could be partially explained by their delayed transcriptomes, since these genes are normally upregulated during early development [50]. Hence, the similar identity of the pathways downregulated in $Ogt^{T931del/Y}$ blastocysts could also suggest a developmental delay. We tested whether DEGs due to the T931del mutation could be explained by a developmental delay by examining their dynamics across preimplantation development in unperturbed embryos. We reasoned that developmental gene expression changes should go in the opposite direction of gene misregulation if the latter is due to a delay. We found that the majority of the most significantly upregulated DEGs as well as many downregulated DEGs cannot be explained by a developmentally delayed transcriptome (S2E Fig), however we cannot exclude this possibility for downregulated genes belonging to mitochondrial or cell adhesion-related GO terms (Fig 2F), which are normally upregulated in the E3.5 trophectoderm. In good agreement, downregulated genes in $Ogt^{T931del/Y}$ blastocysts are overrepresented among genes that in unperturbed embryos are upregulated between the 8-cell stage and the E3.5 trophectoderm (while the same is not significant for ICM; Fig 2F). The downregulation of trophectoderm genes could be linked to a transcriptional delay of the mutant blastocysts or to a more pronounced effect of the lack of O-GlcNAc on this extraembryonic tissue [58].

## Maternal inheritance of a severe hypomorphic *Ogt* mutation perturbs retrotransposons silencing in hemizygous preimplantation embryos

Previous works from us and others showed that the O-GlcNAc modification is required for stable silencing of retrotransposons in mESCs [37,43,59]. However, it remains to be tested whether a milder and physiologically relevant perturbation of O-GlcNAc homeostasis could reactivate specific parasitic promoters *in vivo*. To assess the consequence of reduced OGT's activity on retrotransposon silencing during preimplantation development, we investigated their expression in the $Ogt^{T931del/Y}$ blastocysts. We found that most of the $Ogt^{T931del/Y}$ embryos showed a low magnitude ($\log_2$FC < 1) upregulation of retrotransposons belonging to families of long terminal repeats (LTRs) as well as long interspersed nuclear elements (LINEs; Fig 2G and 2H), while non-autonomous short interspersed nuclear elements (SINEs) contributed less to embryo clustering based on *Ogt* genotype (S2F Fig). Some single repeats were downregulated, but they were generally lowly expressed, and their fold change was lower than upregulated repeats (Figs 2H and S2G). Because of hints of developmental delay of $Ogt^{T931del/Y}$ blastocysts, we tested whether the deregulation of retrotransposons could be explained by an earlier stage's transcriptome. The single repeats which are upregulated in mutant embryos are downregulated between the 8-cell and E3.5 stages in WT embryos; hence, their altered level could be partly due to a developmental delay (S2H Fig). However, the overall negative correlation between developmental expression changes of retrotransposons observed in WT embryos and changes due to the mutation is very weak, implying that a developmental delay is not sufficient to explain retrotransposon upregulation (S2H Fig).

In summary, we found that the maternal inheritance of T931A is sub-lethal with high penetrance already during cleavage stages and characterized the transcriptome of blastocysts with a deletion of T931 which is compatible with preimplantation development. The male $Ogt^{T931del/Y}$

transcriptomes revealed the downregulation of metabolic and cell-adhesion functions, which can be partly explained by a potential subtle developmental delay, and a partial disruption of the silencing of endogenous retroviruses of the ERV1 and ERVK families.

## A mild reduction in OGT's activity affects placental development in a sexually-dimorphic manner

The *Ogt*-Y851A substitution is compatible with life for hemizygous males and homozygous females, although sub-lethality was observed at weaning (Fig 3A). This milder hypomorphic mutant enabled us to study the effect of a prolonged reduction in OGT activity on postimplantation development. We focused on the developing placenta, motivated by our finding that early development is highly sensitive to O-GlcNAc levels and the previous reports that *Ogt* escapes XCI in the extraembryonic ectoderm [28,29], resulting in the double dose of the enzyme in female placentae. *Ogt*'s escape from iXCI has been linked to higher sensitivity of male embryos to maternal stress [33,34] and could have other unexplored sexually-dimorphic developmental functions.

We crossed heterozygous $Ogt^{Y851A/+}$ females with hemizygous $Ogt^{Y851A/Y}$ males and analyzed single placentae dissected at E12.5 (Fig 3B) by western blotting and mRNA-Seq. The western blot detection of $OGT_{Y851A}$ confirms that this mutation does not interfere with OGT protein's folding nor stability (Fig 3C). As for the *Ogt* mRNA in the $Ogt^{T931del}$ mutant blastocysts, we found a compensatory increase in the amount of OGT protein produced in all mutant genotypes analyzed, particularly pronounced in the homozygous $Ogt^{Y851A/Y851A}$ females (Fig 3C and 3D). Notably, the global levels of O-GlcNAc in heterozygous females were comparable to the WT males (Fig 3C and 3D), while WT females have a higher level of O-GlcNAc in this tissue because of bi-allelic *Ogt* expression [33]. As expected, the total O-GlcNAc level was reduced in hemizygous males and homozygous females (Fig 3C and 3D). Male hemizygous and female homozygous displayed inverted proportion of long versus short isoforms of OGA (Fig 3C and 3D). The origin of these closely migrating OGA isoforms is currently not known; their size indicates that they are distinct from the two isoforms previously reported in human [60]. One hypothesis could be that the two OGA bands reflect two states of posttranslational modification.

In agreement with a previous study [33], the transcriptome of the single placentae showed a small fraction of genes expressed in a sexually-dimorphic manner, whereby downregulated DEGs are mostly Y-linked and upregulated DEGs are mostly X-linked and include *Ogt* (S3A Fig). We examined the impact of reduced OGT's catalytic rate on gene expression for both sexes. We found a few hundred DEGs in hemizygous $Ogt^{Y851A/Y}$ male placentae compared to their WT littermates (Fig 3E) and only one in homozygous $Ogt^{Y851A/Y851A}$ females compared to heterozygous ones (Fig 3F). However, DEGs found in hemizygous males were also dysregulated in homozygous females, albeit to a non statistically significant extent (Fig 3G). Female heterozygous and WT males were indistinguishable based on the expression of DEGs (Fig 3G). These observations provide additional support to the dose-dependent gradient of sensitivity to *Ogt* disruption. The lower sensitivity of the homozygous females' transcriptome to *Ogt* disruption (Figs 3E, 3F and S3B) seems difficult to reconcile with their comparable O-GlcNAc level to hemizygous males, namely lower than in WT males and heterozygous female placentae (Fig 3C). We speculate that the western blot technique is not sensitive enough to detect subtle differences in O-GlcNAcylation that could impact gene expression. In the hypothetical case that O-GlcNAc levels were truly identical between *Y851A/Y* and *Y851A/Y851A*, an alternative explanation could be the existence of an X-linked genetic modifier that in females escapes XCI.

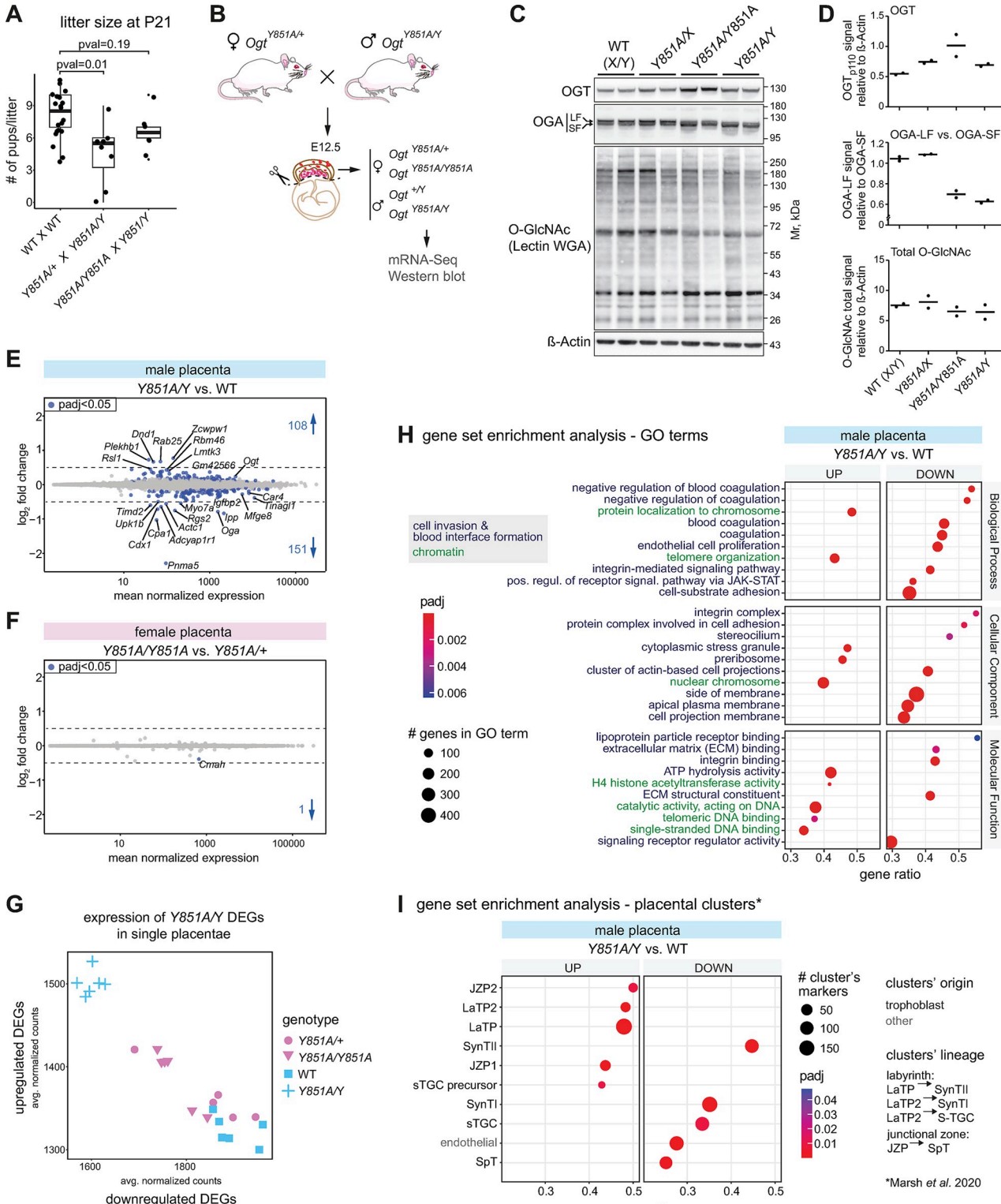

**Fig 3. A mild reduction in OGT's activity affects placental development in a sexually-dimorphic manner.** (**A**) Litter size for intercrosses of WT mice and for crosses between mice bearing the *Ogt*-Y851A mutation. P-values are from unpaired two-sided Student's t-test, assuming unequal variance. N = 20 WT crosses, 8 *Ogt^{Y851A/+}* x *Ogt^{Y851A/Y}* and 6 *Ogt^{Y851A/Y851A}* x *Ogt^{Y851A/Y}* crosses. (**B**) Breeding scheme used to produce placentae with the four different genotypes, which were analyzed via western blot and mRNA-Seq. (**C**) Western blot analysis of OGT, OGA and O-GlcNAc levels in the placenta isolated from E12.5 embryos of the four genotypes produced by the cross described in (**B**). LF: long form, SF: short form. (**D**) Normalized

optical density for OGT (top), O-GlcNAc entire lane (middle) and ratio OGA long form vs. OGA short form from the western blot in (**C**). In (**C,D**), for each genotype, the two placentae belong to embryos from two different litters (the same two litters for all genotypes), except for the hemizygous males which both come from the same litter. (**E,F**) Separate DESeq2 analyses for female and male placentae, comparing (**E**) $Ogt^{Y851A}$-homozygous versus heterozygous female placentae or (**F**) hemizygous versus WT male ones. All genes with adj. p-value < 0.05, any log$_2$FC are colored, and their number is indicated. Genes standing out (and with abs(log2FC) $\geq$ 0.2) are labeled. (**G**) Scatter plot representing the average DESeq2-normalized counts of upregulated and downregulated DEGs found in $Ogt^{Y851A/Y}$ (versus WT male) placentae, in single placenta of all four genotypes analyzed. (**H**) GSEA of gene expression change in hemizygous $Ogt^{Y851A/Y}$ versus WT male placentae. The first 10 GO terms for the three gene ontologies based on Normalized Enrichment Score (NES) are shown. Terms are ordered based on gene ratio. The size of dots is proportional to the number of total genes of a GO term. Gene ratio = fraction of total genes of the GO term which are concordantly changing in mutant embryos. (**I**) Enrichment analysis of placental clusters' marker genes [63] expression changes in $Ogt^{Y851A/Y}$ versus WT male placentae. All enriched clusters (adj. p-value < 0.05) are shown, ordered by gene ratio. The size of dots is proportional to the number of markers of each placental cluster. Gene ratio = fraction of total markers which are concordantly changing in mutant embryos. JZP: junctional zone precursors; LaTP: labyrinth trophoblast progenitors; SynT-I: syncytiotrophoblast layer I; SynT-II: syncytiotrophoblast layer II; sTGC: sinusoidal trophoblast giant cells; SpT: spongiotrophoblasts. In (**E-I**), N per genotype = 6 placentae coming from at least two different litters, except female $Ogt^{Y851A/+}$ placentae for which one sample was excluded because outlier in unsupervised clustering.

The most significant downregulated gene sets in male $Ogt^{Y851A/Y}$ placentae included *blood coagulation*, *endothelial cell proliferation* and many complexes and functions associated with the plasma membrane and the extracellular matrix (Fig 3H). These gene sets point to the process of establishment of the maternal-fetal blood interface that is the placental labyrinth [61]. The E12.5 mouse placental labyrinth consists of two layers of multinucleated syncytiotrophoblasts (SynTI/II) that constitute the maternal-fetal gas and nutrient-exchange surface, and hormone-producing sinusoidal trophoblast giant cells (sTGCs); these three cell types differentiate from common labyrinth trophoblast progenitors (LaTPs) [62]. Between the labyrinth and the maternal decidua, the junctional zone hosts additional cell types with endocrine functions [61]. To test the dysmorphism (or delay) in the formation of the different placental layers, we analyzed the expression of markers for all placental cell types spanning the E9.5 to E14.5 developmental window [63]. Markers for precursor cell types of both labyrinth and junctional zone (LaTPs and JZPs) were all upregulated in $Ogt^{Y851A/Y}$ placentae, while markers for matured cell types (SynTs, sTGCs, SpT), which increase in percentage from E10.5 to E12.5 [63], were all downregulated (Fig 3I). Of note, downregulation was also observed for markers of endothelial cells, which also compose the fetal-maternal exchange barrier but are not of trophoblast origins (Fig 3I). This transcriptional phenotype was highly penetrant among $Ogt^{Y851A/Y}$ embryos, and again not as evident in heterozygous or homozygous females (S3B Fig). Taking together the data above, we conclude that males with a reduced OGT activity display a delay in the differentiation of a functional embryonic placenta. This delay could be tissue-specific, for example due to $Ogt$'s role in the protein networks establishing cell-matrix contacts, or it could be associated with a slight developmental delay of the whole hemizygous $Ogt^{Y851A/Y}$ embryo. We tested the latter possibility by staging $Ogt^{Y851A/Y}$ embryos by morphometric analysis of limb development [64]. This method has a precision of 0.5 day around E12.5 and did not reveal a difference between $Ogt^{Y851A/Y}$ and WT embryos (S3C Fig). However, we cannot exclude a delay of less than 12-hours.

Next, we assessed the impact of the *Y851A* mutation on the silencing of retrotransposons in the placenta. Hemizygous $Ogt^{Y851A/Y}$ placentae but not homozygous $Ogt^{Y851A/Y851A}$ showed a very low magnitude upregulation of mRNAs transcribed from promoters of LTR and LINE classes of retrotransposons: LTRs of the endogenous retrovirus type K (ERVK) (*IAPLTR2_Mm*), two evolutionary recent LINE1 (*L1MdTf_I* and *L1MdTf_II*) and a full-length murine endogenous retrovirus-L (*MERVL-int*; S3D Fig). These classes of retrotransposons are capable of retro-insertion [65] but the internal part (e.g. *Gag*, *Pol*) of IAPs, the most active autonomous LTR elements, was not significantly overexpressed in $Ogt^{Y851A/Y}$ placentae (S3E Fig). Thus, the epigenetic silencing mechanism that represses autonomous retrotransposons is

overall maintained upon a 30% reduction in OGT's catalytic rate and *IAPLTR2_Mm* containing transcripts are likely produced by solo LTRs. We also observed a significant upregulation of two types of satellite repeats (IMPB_01 and MMSAT4), which are interspersed in the genome and very scarcely described in literature (S3F Fig). It is noteworthy that male *Ogt*$^{Y851A/Y}$ placentae additionally upregulated gene sets related to chromatin remodeling and DNA binding (Fig 3H). Heterochromatin being essential to repress LINE1, LTRs and satellites [66], a partial heterochromatin disruption could explain the increase in retrotransposons' transcripts in the *Ogt*-mutant male placenta.

In conclusion, the data showed that a modest reduction in OGT's catalytic rate is sufficient to cause embryonic sub-lethality associated with male-specific delayed or defective placentation.

## Progressive dampening of cellular activities after acute degradation of endogenous OGT *in vitro*

To rapidly deplete OGT protein *in vivo*, we implemented an OGT-degron that utilizes two knock-in alleles: i. the insertion of the *Oryza sativa TIR1* gene (*OsTIR1*) downstream of the ubiquitous promoter of the ROSA26 locus [67] (ROSA26$^{OsTIR}$ allele, Fig 4A) and ii. the N-terminal insertion of the minimal version of the AID peptide (44 amino acids) [68,69] in frame with the longest isoform of *Ogt* (known as nucleocytoplasmic *Ogt* or *ncOgt*; *Ogt*$^{NterAID-MYC-FLAG}$ allele, henceforth *Ogt*$^{AID}$, Fig 4B).

Heterozygous *Ogt*$^{AID/WT}$ females were viable at birth, healthy, fertile and transmitted the mutation at the expected Mendelian ratio. However, hemizygous mutant males *Ogt*$^{AID/Y}$ were never found at weaning, indicating that the addition of the AID tag to endogenous OGT impairs its function *in vivo*. *Ogt*$^{AID/Y}$ males were viable at E7.5 and expressed the AID-tagged OGT (E7.5; S4A Fig), therefore the *Ogt*$^{AID}$ allele is suitable to address OGT's function at early embryonic stages. We then tested the efficiency of the AID-OGT degron system in mouse embryonic fibroblasts (MEFs) derived from the breeding of females *Ogt*$^{AID/WT}$ with males homozygous for the *OsTIR* transgene (S4B Fig). We analyzed the male *Ogt*-hemizygous clones, which only express either the WT or the AID-OGT protein. Of note, the presence of the AID tag caused a reduction of OGT protein amount of about half without auxin treatment (Figs 4C and S4C), which could explain the lethality observed in hemizygous males. Upon the addition of auxin to the culture medium, the level of AID-OGT dropped to about 20% of that of WT after 30' minutes (Figs 4C and S4C). This reduction persisted for at least 4 days (S4D Fig) and resulted in a global decrease of O-GlcNAc levels from 24 hours onwards (S4E Fig). Having validated the OGT-degron system in MEFs, we attempted to deplete the maternal payload of OGT in preimplantation embryo grown *ex vivo*. However, in this model the AID-induced O-GlcNAc perturbation was suboptimal and likely necessitates alternative perturbation strategies (S5 Fig and S1 Text).

Next, we used the OGT-degron system in MEFs to gain insights into the immediate transcriptional response after OGT acute depletion and the kinetic of gene expression changes. We performed mRNA-Seq of three different clones of *OsTIR,Ogt*$^{AID}$ MEFs treated with auxin for 24, 48 and 96 hours and compared them with the same clones cultured in parallel in the absence of auxin (Fig 4D). Because the addition of the AID tag halves OGT protein amount (Figs 4C and S4C), we also compared *OsTIR,Ogt*$^{AID}$ versus *OsTIR,Ogt*$^{WT}$ in the absence of auxin and found about one hundred DEGs (S6A Fig, left), mainly involved in actin-based cell motility, a homeostatic program of fibroblasts (Fig 4E). Additionally, the side effect of auxin was controlled by treating clones devoid of the AID tag (*OsTIR,Ogt*$^{WT}$). We found a small but not negligible gene expression change solely caused by the drug (S6A Fig, right) and involving

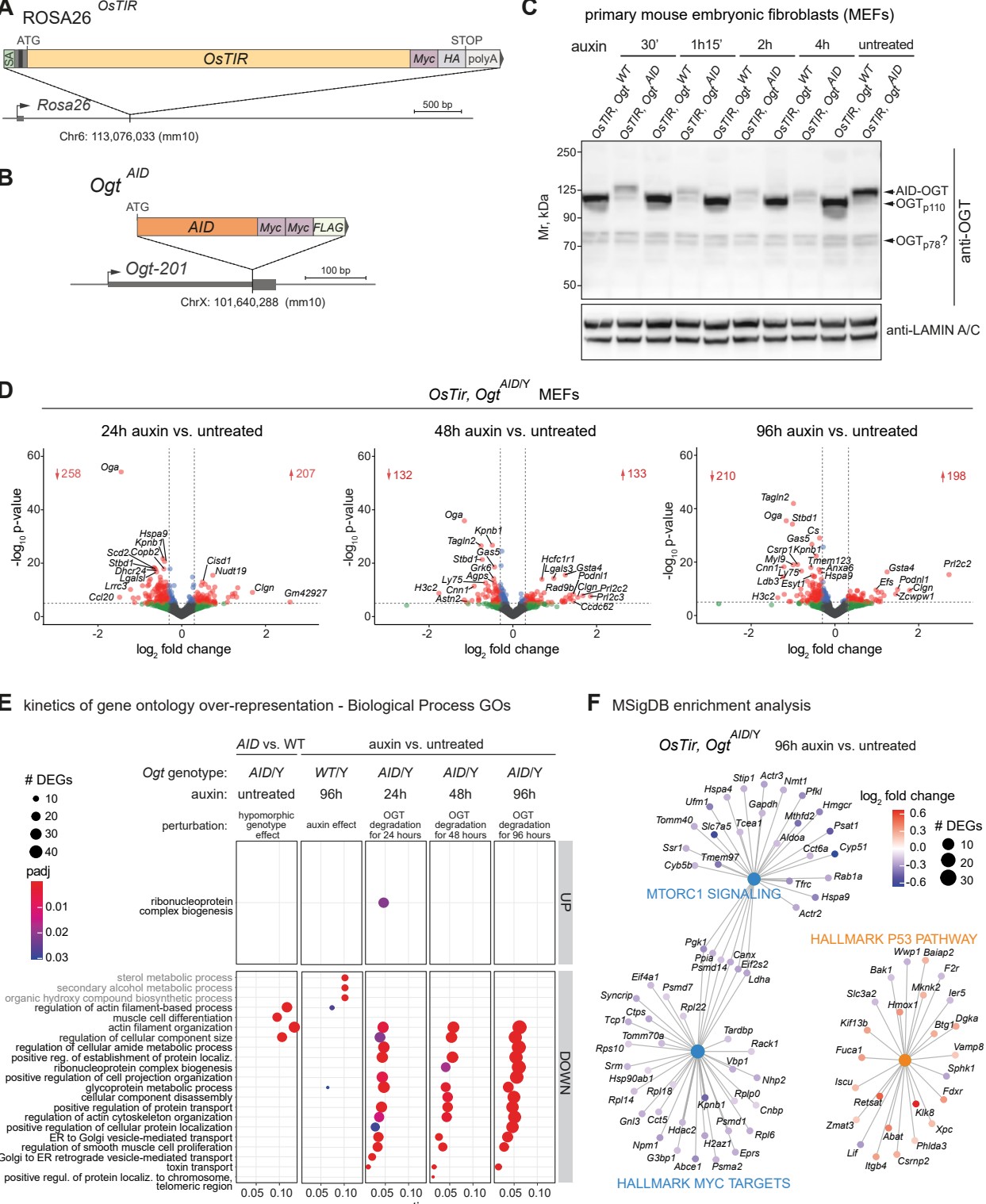

**Fig 4. Kinetics of differential gene expression after rapid degradation of endogenous OGT in MEFs. (A)** The ROSA26*OsTIR* allele was created by insertion of the coding sequence of *Oryza sativa* TIR1 gene (*OsTIR1*; fused to C-terminal Myc and HA adjacent tags) downstream the ubiquitous ROSA26 promoter. The transgene is integrated at the splicing acceptor (SA) site of ROSA26, resulting in ROSA26's interruption and expression of *OsTIR1* from the endogenous promoter. **(B)** The *Ogt*^*AID*^ allele consists of the targeted insertion of the minimal AID peptide's sequence (44 amino acids) immediately downstream the initiating ATG codon of the longest *Ogt* isoform (*Ogt-201* or ENSMUST00000044475 or *ncOgt*), in frame with

two Myc and one FLAG epitope tags. (**C**) Western blot of endogenous OGT (ab177941 antibody) in whole cell protein extracts from male primary MEFs, untreated or treated with auxin for different amounts of time. From the same litter, an $OsTIR,Ogt^{AID}$ clone and a control $OsTIR,Ogt^{WT}$ clone were treated in parallel. Lamin A/C was probed as a loading control. (**D**) Volcano plots from DESeq2 analysis of gene expression changes in $OsTIR,$ $Ogt^{AID}$ clones after 24, 48 and 96 hours of auxin treatment compared to untreated $OsTIR,Ogt^{AID}$ clones grown in parallel. Differentially expressed genes with p-value $< 10^{-5}$ and absolute $\log_2$FC $> 0.3$ (i.e. 1.2-fold increase or decrease in expression) are colored in red and their number is indicated. (**E**) Gene ontology (GO) over-representation analysis of DEGs (adj. p-value $< 0.05$, any $\log_2$FC) from the same comparisons as in (**D**), plus untreated $OsTIR,Ogt^{AID}$ versus untreated $OsTIR,Ogt^{WT}$ clones (effect of the hypomorphic genotype), and $OsTIR,Ogt^{WT}$ control clones treated with auxin for 96 hours versus grown untreated (effect of the auxin drug). The first 25 most enriched Biological Process GO terms are shown, based on p-value across all comparisons. Terms are ordered by gene ratio. Gene ratio = genes belonging to the GO term / total number of deregulated DEGs for that comparison. UP = upregulated DEGs, DOWN = downregulated DEGs. Terms enriched due to auxin treatment on control clones are written in gray. (**F**) Analysis of enrichment for Hallmark gene sets of the Molecular Signatures Database (MSigDB) among DEGs after 96 hours of acute OGT depletion. For (**E-F**), DEGs after acute OGT depletion which were also differentially expressed in auxin-treated control cells were excluded before enrichment analyses of auxin-treated $Ogt^{AID}$ clones; DEGs with mean DESeq2-normalized counts $< 10$ across all samples were excluded from all enrichment analyses.

sterol metabolism (Fig 4E). All DEGs appearing upon auxin addition in control $OsTIR,Ogt^{WT}$ clones were excluded from the gene ontology (GO) analysis of all the other comparisons.

Acute depletion of endogenous OGT significantly affected the expression of hundreds of genes, with most of the $\log_2$FC below 1 in both directions (Fig 4D). As expected, we found *Oga* transcripts' downregulation and the reciprocal *Ogt*'s upregulation among the most significant changes (Figs 4D and S6B) [52–57]. The DEGs significantly overlapped between the three time points (Fisher exact test's p-value $\ll 1e^{-100}$ for all comparisons between two time points; S6C Fig). A few gene ontology terms were enriched among upregulated DEGs and only at the earliest 24 hour time point, indicating an acute response to OGT or O-GlcNAc depletion leading to an increase in transcripts encoding proteasomal subunits and proteins involved in ribosome biogenesis and mitochondrial translation (Figs 4E and S6D). Instead, numerous terms were consistently downregulated after OGT depletion, with statistical significance increasing from 24 to 96 hours of acute OGT depletion (Figs 4E and S6D). The most affected cellular functions at all time points were the mesenchymal transcriptional program, including the actomyosin machinery (e.g. *Cnn1*, *Myl9*) and transforming growth factor beta 3 (*Tgfb3*) [70], and intracellular membrane trafficking. Additionally, after 48 hours of OGT's depletion, proteasomal subunits and proteins involved in ribosome biogenesis started to be downregulated (Figs 4E and S6D), although other genes than the ones upregulated at the 24 hour time point. Finally, at the latest time points, genes involved in translation were more broadly impacted (S6D Fig). Notably, it was reported that the loss of non-catalytic functions versus the loss of the O-GlcNAc transferase activity of OGT alters the abundance of different sets of proteins, including the increase in mitochondrial translation-related proteins in the absence of non-catalytic functions and the downregulation of translation-associated factors in the absence of O-GlcNAcylation [18]. Hence, it can be hypothesized that the transcriptional response observed at the earliest time point has a higher contribution from the abrupt depletion of OGT molecules (Fig 4C), while the response accumulating over time is due to the following decrease in O-GlcNAc (S4E Fig).

The gene network analysis showed that downregulated genes were enriched for targets of the transcription factor c-Myc and of the mechanistic target of rapamycin kinase 1 (mTORC1), starting from 48 hours and 96 hours of auxin treatment, respectively (Fig 4F). *c-Myc* is an oncogene stimulating cell growth [71], while mTORC1 senses cellular nutrients' and growth factors' availability to balance between anabolism and macromolecule recycling [72]. Downregulated c-Myc targets include proteasomal subunits, ribosomal proteins, translation initiation factors and *Hdac2* (Fig 4F). The dampening of these two growth pathways was accompanied by signs of activation of a p53 stress response (Fig 4F), although there was no significant enrichment of gene networks associated with apoptosis or cellular senescence. p53

signaling can repress translation via multiple post-translational and transcriptional mechanisms, including interfering with c-Myc's activity [73], thus the later-appearing downregulation of protein synthesis-related genes could be a consequence of a primary stress response to the lack of OGT. On the other hand, c-Myc has been reported to be O-GlcNAcylated [74] and a recent study found that the O-GlcNAcylation of a regulator of mTORC1, known as Raptor, signals glucose status [75], thus O-GlcNAc could also act directly on these pathways. Our experiment does not allow establishing if a relationship of causality exists between the stress response and the dampening of anabolic transcription caused by OGT reduction.

## Discussion

While it has been known for over two decades that maternal *Ogt* is essential for embryonic survival [15,16], it remained unclear whether non-catalytic roles of OGT could contribute to the lethal phenotype. In particular, OGT's N-terminus composed by 13 tetratricopeptide (TPR) repeats mediates interactions with various proteins [76] and could potentially also play an essential structural role. Our finding that the embryonic survival of the catalytic mutants mirrors the *in vitro* decrease in OGT's glycosyltransferase catalytic rate indicates that the observed phenotypes are caused by the gradual impairment of O-GlcNAcylation. Thus, we provide here the first demonstration that O-GlcNAc homeostasis is necessary for mouse embryonic development.

OGT's glycosyltransferase catalytic core is endowed with a second enzymatic activity: the proteolytic cleavage of the cell cycle regulator HCF-1 [6,77], whose maturation is proposed to ensure a proper progression through S phase and cytokinesis [78]. However, while OGT's O-GlcNAc transferase activity was found to be necessary for the proliferation of cycling cells, it does not seem to be the case for the cleavage of HCF-1 by OGT [18]. In addition, the H568A mutation (the most severe hypomorphic allele of this study, disrupting completely the glycosyltransferase activity) does not affect HCF-1 proteolysis [77].

It is very difficult to identify the O-GlcNAc modified proteins whose de-O-GlcNAcylation causes embryonic lethality. The observed molecular and developmental phenotypes could arise from the cumulative effect of the de-O-GlcNAcylation of multiple sites/proteins. Cell cycle defects could be involved given the requirement of O-GlcNAcylation for cell proliferation, although we did not observe the activation of cell cycle checkpoints in our transcriptomics data. Recently, mitochondrial dysfunction triggered by mTOR hyperactivation was proposed to be the cause of cellular lethality following *Ogt*-KO in mESCs [17]. We did not find signs of oxidative stress in the transcriptomes of *Ogt*-hypomorphic embryos; however, this remains a possibility that would require to be tested through assays of mitochondrial function.

We analyzed the hypomorphic mutants for which we could obtain viable hemizygous male embryos. The effect of the most severe perturbation of O-GlcNAc homeostasis ($Ogt^{T931A/del}$) on the blastocyst's transcriptome suggests a mild developmental delay. The deregulated transcriptome of E12.5 placentae from the milder hypomorphic mutant $Ogt^{Y851A}$ also points to the same direction. The postimplantation sub-lethality of the $Ogt^{Y851A}$ mutants is consistent with a threshold effect lying in insufficient growth support; thus placental dysfunction might have a greater contribution than defects in the embryo proper to embryonic lethality caused by milder reductions of O-GlcNAc.

Because of OGT's double dose in the female extraembryonic ectoderm [28], another consideration is that the ExE and the deriving placenta are good candidates to be responsible for the observed higher peri-/postimplantation mortality of males upon *Ogt*'s dysfunction. Remarkably, the double dose of *Ogt* in female placentae is conserved in humans [33], motivating future studies on the sexually-dimorphic role of *Ogt* in human. For example, it would be interesting to investigate the contribution of placental dysfunction to OGT-XLID neurodevelopmental

syndrome. Of note, male-specific higher sensitivity to maternal stress due to lower placental OGT dose has been linked to neuronal effects, namely hypothalamic-pituitary-adrenal stress axis hyperresponsivity as well as changes in hypothalamic gene expression [33,34].

Rapid hypo-O-GlcNAcylation in MEFs caused the dampening of various basal cellular activities, including translation. Again, due to the numerous *Ogt* targets, it is likely that O-GlcNAc acts on growth through multiple pathways. Thus, it is difficult to pinpoint the exact ones involved in the observed transcriptional response. Nonetheless, it is worth noticing that some cellular functions are consistently downregulated in the two models (MEFs and embryo): intracellular membrane trafficking as well as actin-related processes. Moreover, O-GlcNAc-depleted embryos show evidence of a stress response involving translation, specifically the increase in stress granules. This result is consistent with the regulation of stress granules' assembly by the O-GlcNAcylation of YTHDF1/3 proteins [79].

The other outstanding molecular phenotype observed in multiple datasets analyzed is the disturbance of chromatin-based gene silencing upon O-GlcNAc perturbation: hypomorphic *Ogt* mutants revealed that retrotransposons upregulation occurs *in vivo* when O-GlcNAc homeostasis is perturbed, even moderately. The target responsible for this phenotype remains to be elucidated. The extent of upregulation of retrotransposons in hypo-O-GlcNAcylated blastocysts is modest in comparison to that caused by inactivating mutations of *Trim28* [37,80] or global DNA demethylation [81]. Hence, our results could indicate that the residual level of O-GlcNAc contributes to retrotransposon silencing. Alternatively, hypo-O-GlcNAcy-lation could cause a global and incomplete disruption of heterochromatin rather than severely impacting the repressors of retrotransposons (e.g. HUSH and TRIM28 complexes or DNA methylation maintenance). This hypothesis is supported by a recent study showing that *Ogt*-KO in mESCs causes a genome-wide partial loss of 5-meythylcytosine genome-wide and that the association between OGT and TET proteins is the likely responsible player [43].

Our comparative study of different degrees of impairment of OGT's activity set the thresh-olds for inheritance and hemizygous embryonic lethality. We found that the $Ogt^{Q449N}$ allele, which drastically reduces the enzymatic activity to 2% of the WT, causes lethality to both male and female embryos. In contrast, the milder reduction of OGT's enzymatic activity to 18% of that of the WT caused by the *Ogt*-T931A substitution resulted in low frequency of maternal transmission of this allele and was embryonic lethal for hemizygous. The mildest reduction of OGT's activity tested (*Ogt*-Y851A, 24% as compared to WT) is compatible with maternal inheritance and life for both homozygous and hemizygous. To our knowledge, Y851A is the second reported murine *Ogt* mutation that perturbs O-GlcNAc homeostasis while being com-patible with adult life and reproduction, the other being *Ogt*-C921Y, which is a human patho-logic variant causing OGT-XLID. Based on structural data, the role of C921 in OGT's catalytic mechanism is more difficult to predict than for the mutations characterized in this work [46]. Although the C921Y substitution was shown to reduce OGT $V_{MAX}$ *in vitro* to 25% of the WT [82], these mutant mice do not show embryonic sub-lethality, but present features shared with XLID human patients, making this model very well suited to study XLID's pathology [83]. The $Ogt^{Y851A}$ allele created in this study should be useful for future investigations of the conse-quences of prolonged reduction of O-GlcNAcylation in development and adulthood, in physi-ology and disease mechanisms relevant for human health, such as diabete mellitus [25,84].

## Materials and Methods

### Ethics statement

Animal husbandry and experimentation was reviewed, approved, and monitored under the European Molecular Biology Laboratory (EMBL) Institutional Animal Care and Use

Committee (protocol number 21–012_RM_MB) and the Italian Ministry of Health (protocol numbers 17/2019-PR to C.L. and 598/2023-PR to M.B.).

## Animal care and handling

Mice were housed in the pathogen-free Animal Care Facility at EMBL Rome on a 12-hours light-dark cycle in temperature and humidity-controlled conditions with *ad libitum* access to food and water.

## Generation of murine *Ogt* allelic series: $Ogt^{Y851A}$, $Ogt^{T931A}$, $Ogt^{Q949N}$, $Ogt^{H568A}$, $Ogt^{NLS-}$, $Ogt^{NterAID-MYC-FLAG}$

*Ogt* mutant alleles were created by CRISPR/Cas9-editing in the zygote (FVB/NCrl genetic background, Charles River) as previously described [85]. Transgenic mouse production was performed by the Gene Editing and Virus Facility at EMBL Rome. Sequences of the single-stranded donor templates are provided in S1 Table.

Briefly, for $Ogt^{Y851A}$ (FVB/NCrl-$Ogt^{em1(Y851A)}$Emr) CRISPR crRNA oligo (TAGATGG-GACGTCTACACCC) was annealed with tracrRNA and combined with a homology flanked ssODN donor coding for Tyrosine at amino acid position 851, substituting the wild type Alanine. The target location was exon 19 of transcript *Ogt-201*, (ENSEMBL v109); genomic coordinate: ChrX. 100719847–100719886 (GRCm39/mm39).

Similarly, for $Ogt^{T931A}$ (FVB/NCrl-$Ogt^{em2(T931A)}$Emr) crRNA oligo (TTGTGTAATGGAC ACACCAC) was combined with a ssODN donor coding for Threonine at amino acid position 931 substituting Alanine, exon 20, transcript *Ogt-201*, ChrX. 100722515–100722518.

For $Ogt^{Q949N}$ (FVB/NCrl-$Ogt^{em3(Q949N)}$Emr) crRNA oligo (TAGATGGGACGTCTACACC C) was combined with a ssODN donor coding for Glutamine at amino acid position 949 substituting Asparagine, exon 19, transcript *Ogt-201*, ChrX. 100719847–100719886.

For $Ogt^{H568A}$ (FVB/NCrl-$Ogt^{em4(H568A)}$Emr) crRNA oligos (GCTATGTGAGTTCTGACT TC and ATGAAGTGTGGAATACGTCA) were combined with a ssODN donor coding for Histidine at amino acid position 568 substituting Alanine, exon 13, transcript *Ogt-201*, ChrX. 100713458->100713464.

For $Ogt^{NLS-}$ (FVB/NCrl-$Ogt^{em5(DFPmut)}$Emr) crRNA oligos (CAATAAGCATCAGGAAA GTC and GCCAAGTTACAATAAGCATC) were combined with a ssODN donor coding for three Alanines at amino acid positions 461–463 substituting three amino acids associated with *Ogt* NLS, Aspartic acid, Phenylalanine and Proline, exon 11, transcript *Ogt-201*, ChrX. 100713458->100713464.

For $Ogt^{NterAID-MYC-FLAG}$ (FVB/NCrl-$Ogt^{em6(AID)}$Emr) crRNA oligos (CTCCAGATGGC GTCTTCCGT and ACTGTCGGCCACGTTGCCCA) were combined with a ssDNA donor coding for AID, 2xMyc and 1xFLAG tags inserted at *Ogt* N-terminus, transcript *Ogt-201*, ChrX. 100683892.

In all cases, annealed sgRNAs were complexed with Cas9 protein and combined with their respective ssDNA donors (Cas9 protein 50 ng/μL, sgRNA 20 ng/μL, ssDNA 20 ng/μL. All single stranded DNA and RNA oligos were synthesized by IDT. Cas9 protein (IDT), sgRNA and ssDNA donor template were co-microinjected into zygote pronuclei using standard protocols [86] and after overnight culture 2-cell embryos were surgically implanted into the oviduct of day 0.5 post-coitum pseudopregnant CD1 mice.

Founder mice were screened for the presence of the mutation by PCR, first using primers flanking the sgRNA cut sites which identify InDels generated by NHEJ repair and can also detect larger products implying HDR. Secondly, 5' and 3' PCRs using the same primers in

combination with template-specific primers allowed for the identification of potential founders. These PCR products were then Sanger sequenced and aligned with the *in silico* design.

The *Ogt* mutant strains were maintained on an FVB/NCrl genetic background. PCR genotyping using gDNA isolated from tail biopsies was performed at weaning (postnatal day 21), using primers listed in S2 Table.

## Generation of the *ROSA26^OsTIR* allele

The *OsTIR1* cDNA was modified by the addition of Myc and HA tags for protein detection [87]. The *OsTIR1-Myc-HA* transgene was next inserted in a targeting vector specific to the ROSA26 locus. A *lox-stop-lox* (*LSL*) cassette was added before the transgene to prevent protein expression. The *LSL-OsTIR1* targeting construct was electroporated into A9 ESCs (129xC57BL/6 genetic background) [88]. Southern blotting of the individual ESC-clone-derived DNA was used to identify homologous recombinants. A9 ESCs with successful insertion were injected into C57BL/6 8-cell stage embryos that were then transferred into foster mothers as previously described [88]. The resulting chimeras were tested for germline transmission. $F_1$ mice bearing the *OsTIR1-Myc-HA* transgene were crossed to the FLP-expressing transgenic mice (FLPeR) [89] to remove the frt-flanked neo cassette. The resulting progeny was crossed for 12 generations with C57BL/6N animals to obtain the C57BL/6N-ROSA26^tm1.1(osTIR1)ChLan mouse line (ROSA26-*LSL-OsTIR1*). To remove the *LSL* cassette, the ROSA26-*LSL-OsTIR1* mice were crossed to the Cre-deleter strain [90], generating the FVB;B6J;129-Gt(ROSA)26Sor^tm1(OsTIR)Emr allele (hereafter ROSA26^OsTIR), constitutively expressing *OsTIR1*. The ROSA26^OsTIR allele was then backcrossed on FVB/NCrl genetic background for 2 generations.

## Dissection of mouse placentae

Embryos were dissected at E12.5 post-coitum from the uteri of naturally mated mice. Placentae were dissected in phosphate-buffered saline solution (PBS) + 0.1% Bovine Serum Albumin (BSA) and the exterior decidua was peeled out to minimize the presence of maternal tissue. For RNA-Seq, clean placentae were bisected via transverse sectioning; one half was collected in a 1.5 mL tube containing 300 µL of 1x RNA Protection Reagent (NEB #T2011-1) and snap-frozen. The E12.5 whole embryos were cleaned from the yolk sac and imaged with trans-illumination on Leica M205C dissection microscope. For limb staging, the images were analyzed using the eMOSS software (https://limbstaging.embl.es/) [64].

## Derivation and culture of mouse embryonic fibroblasts (MEFs)

Primary MEFs were derived from E12.5 post-coitum embryos as previously described [91]. Briefly, the head and the organs were removed, and gDNA was extracted from the head using the PCRBIO Rapid Extract Lysis Kit (PCR Biosystems #PB15.11–24) for PCR genotyping with primers described in S3 Table. The embryonic body was minced using sterile forceps and small pieces of tissue were transferred to 5 mL Trypsin-EDTA (0.05%; Gibco #25300054) and homogenized through repeated pipetting. The homogenized tissue was spun down, the supernatant was removed and the dissociated cells resuspended in MEF medium (Dulbecco's Modified Eagle Medium (DMEM; Gibco #41965–039), 10% fetal bovine serum (PANBiotech #3306-P131004), 100 U/mL Penicillin/Streptomycin (Gibco #15140122), 1x GlutaMAX (Gibco #35050061)). Cells were cultured on 0.1% gelatin in an incubator at 5% $CO_2$ and 37°C. MEFs were passaged at ~80% confluency using Trypsin-EDTA (0.05%) for cell detachment and dissociation. All experiments were performed between passage 1 and 4. The three pairs of

clones studied were derived from three different litters, each pair (*OsTIR,Ogt^{AID}* and control *OsTIR,Ogt^{WT}*) coming from the same litter.

## Auxin treatment of MEFs

Indole-3-acetic acid sodium salt, or IAA or auxin (Sigma-Aldrich #I5148), was resuspended in ddH$_2$O under sterile conditions to a stock concentration of 10 mM, aliquoted and stored at -20˚C. For auxin treatment of MEFs, 500 μM of auxin was added to the culture medium at day 0 and cells were collected 24, 48 and 96 hours later. For the 96-hour time point, the medium was changed after 48 hours with fresh MEF medium supplemented with auxin 500 μM, to keep auxin concentration constant throughout the experimental time. At the time of collection, 500 μM auxin was added to all reagents used for cell dissociation (Trypsin-EDTA and DPBS). Cells were collected in 1.5 mL tubes, resuspended in 500 μL of DPBS (Gibco #14190–094) with or w/o 500 μM auxin and the same volume of 2x Monarch DNA/RNA Protection Reagent (NEB #T2011-1) was added before snap-freezing and storage at -80˚C until RNA extraction.

## Collection of embryos from natural mating

Superovulation of 6–12 weeks-old FVB females was induced by hormonal stimulation (5 IU of PMSG and 5 IU of hCG 64 hr and 16 hr before collection, respectively) and cumulus-oocyte complexes were collected in warm M2 medium (Sigma-Aldrich #M7167), then moved to a drop of warm hyaluronidase (Sigma-Aldrich #H4272-30 mg). Individual oocytes were collected, washed 4–5 times in M2 and then moved to culture.

## In vitro fertilization

In vitro fertilization (IVF) was performed based on the published protocol, with minor modifications [92]. When WT sperm was used for IVF experiments, the strain was always FVB/NCrl except for the *Ogt^{T931A}* blastocysts Smart-Seq experiment where it was PWD/PhjEmr. Superovulation of 6–12 weeks-old FVB females was induced by hormonal stimulation (5 IU of PMSG (Prospec Bio #HOR-272) and 5 IU of hCG (Sigma-Aldrich #CG5-1VL) 64 hours and 16 hours before collection, respectively), and cumulus-oocyte complexes were collected in KSOM medium containing GSH (final concentration 10 mM; Sigma-Aldrich #G6013). Concomitantly, cauda epididymis and vasa deferentia from FVB or PWD males were dissected and sperm was gently squeezed out into capacitation medium (HTF supplemented with MBCS at final concentration of 0.75 mM; HTF: Sigma-Aldrich #MR-070-D, MBCD: Sigma-Aldrich #C4555) and allowed to swim up for 1 hour. Sperm were subsequently counted and cumulus-oocyte complexes were inseminated with 0.2 million sperm in a 200 μL fertilization drop. Four hours after sperm addition, zygotes were cleaned from the surrounding cumulus cells and sperm by 5–6 washes in KSOM, then transferred to culture media. Embryos were cultured in a standard mammalian cell incubator (37˚C, 5% CO$_2$) in EmbryoMax KSOM Mouse Embryo Medium (Sigma-Aldrich #MR-106-D).

## Auxin treatment of embryos grown *ex vivo*

For auxin treatment of zygotes derived from IVF, 250 μM of auxin (and 500 μM for OGT staining 3.5 hours post-IVF) was added to the fertilization drop (KSOM medium containing GSH), to all KSOM washing drops and to the KSOM medium used for culture for the next following days. In order to maintain a constant auxin concentration, embryos were moved to fresh KSOM supplemented with 250 μM auxin every second day. For auxin treatment of the

morulae, 72 hours after collection from natural mating, embryos were moved to fresh media either not containing or containing 250 µM of auxin for culture for the next 24 hours.

On the day of fixation for immunofluorescence staining or collection for Smart-Seq, M2 medium was supplemented with 250 µM (or 500 µM) auxin. For Smart-Seq, groups of 5–10 blastocysts were washed twice in warm M2 drops and collected in 4 uL of a lysis buffer containing 0.5% Triton X-100 in $H_2O$, 1 U/µL SUPERase•In RNase Inhibitor (ThermoFisher #AM2696), 2.5 mM dNTPs (BiotechRabbit #BR0600204, 10 mM each) and 2.5 µM oligo-dT primer (5′–AAGCAGTGGTATCAACGCAGAGTACT30VN-3′). Blastocysts were collected in five different replicates of breeding followed by auxin treatment and stored at -80˚C prior to Smart-Seq library preparation and next generation sequencing.

## Immunofluorescence staining of preimplantation embryos

Zygotes (3.5 hours post-IVF) and morulae (70 hours post-IVF) were treated the same way, as previously described in Bošković *et al.* [93] with some minor modifications as follows. After two washes in M2 medium (Sigma-Aldrich #M7167), the zona pellucida was removed by a brief incubation in drops of warm Acidic Tyrode's solution (Sigma-Aldrich #MR-004-D), followed by other two M2 washes to neutralize the acid. The embryos were then washed once in PBS + 0.1% BSA (Sigma-Aldrich #A2153), fixed in 4% paraformaldehyde (PFA) in PBS for 20' at 37˚C, permeabilized in 0.5% Triton X-100 for 20' at 37˚C and washed three times in PBS-T (0.15% Tween-20 in PBS). The epitope was then unmasked in 50 mM $NH_4Cl$ solution in $H_2O$ for 10' at room temperature, followed by two additional PBS-T washes, and then blocked for 3 hr at room temperature or overnight at 4˚C in BSA (Sigma-Aldrich A2153) 3% in PBS-T. Primary antibody incubation was performed overnight at 4˚C in the blocking solution, followed by three washes in PBS-T, re-blocking for 30' at room temperature, three additional PBS-T washes, secondary antibody incubation for 1 to 2 hr at room temperature in the blocking solution and three final PBS-T washes. The embryos were immediately mounted on coverslips in Vectashield (Vector Laboratories #H-1200) containing 4'-6-Diamidino-2-phenylindole (DAPI) or, in the case of the morulae, in drops of 75% Vectashield in PBS in order to preserve the 3D structure. Fixed immunostained samples were imaged using a Nikon AX scanning confocal (using galvanometric mirrors).

The primary antibodies used were: anti-OGT (Abcam #ab177941), anti-O-GlcNAc clone RL2 (Abcam #ab2739 and Merck Millipore #MABS157). Dilution of all primary antibodies was 1:200. Secondary antibodies used were: A647-conjugated goat anti-rabbit IgG (ThermoFisher #A21244) and A647-conjugated goat anti-mouse IgG (ThermoFisher #A21236). Dilution of all secondary antibodies was 1:500.

## Collection of $Ogt^{T931A}$ blastocysts and RNA extraction

Ninety two hours post-IVF, groups of 5–10 embryos from $Ogt^{T931A/+}$ or $Ogt^{+/+}/Ogt^{T931del/+}$ mothers were washed twice in warm M2 drops. Single blastocysts were collected from the M2 drop in 5 µL of 1x TCL lysis buffer (Qiagen #1031586) containing 1% (v/v) 2-mercaptoethanol (Gibco #31350010) and 0.5 U/µL of SUPERase•In RNase Inhibitor (ThermoFisher #AM2694).

Total RNA was purified using RNAClean XP beads (Beckman Coulter #A63987) according to the manufacturer's protocol for a 96-well plate and Small Volume Reactions. After the last ethanol wash, the RNA was eluted from RNA beads in 8 µL of $H_2O$, 3 µL of which were transferred to a new 96-well plate containing 1 µL of dNTPs (BiotechRabbit #BR0600204, 10 mM each) and 1 µL of 10 µM oligo-dT primer (5′–AAGCAGTGGTATCAACGCAGAG-TACT30VN-3′). The plate was stored at -20˚C prior to Smart-Seq library preparation and next generation sequencing.

Single blastocysts from $Ogt^{T931A/+}$ mothers were genotyped using cDNA. The cDNA from single blastocysts was prepared from 3 μL of RNA (extracted as described above) using Super-Scrip IV RT (ThermoFisher #18090200) according to the manufacturer's instructions and using random hexamers. PCR genotyping of the resulting cDNA was performed with primers in S4 Table and the result was confirmed with Sanger sequencing of the PCR product.

## Smart-Seq library preparation and sequencing from single embryo

Single-embryo full-length cDNA libraries for mRNA sequencing were prepared by EMBL Genomics Core Facility using a modified Smart-Seq2 protocol [51] utilizing SuperScript IV Reverse Transcriptase (ThermoFisher #18090200) and the tagmentation procedure previously described [94]. The retrotranscription reaction mix was as follows: 2 μL SSRT IV 5x buffer, 0.5 μL 100 mM DTT, 2 μL 5 M betaine, 0.1 μL 1 M $MgCl_2$, 0.25 μL 40 U/μL RNAse inhibitor (Takara #2313A), 0.25 μL SSRT IV, 0.1 μL 100 μM TSO, 1.15 μL RNase-free $H_2O$; with thermal conditions: 52˚C 15', 80˚C 10'. cDNA was amplified using 18 PCR cycles. The cDNA cleanup (0.6x SPRI beads ratio; Beckman Coulter #B23319) was carried out omitting the ethanol wash steps and the elution volume was 13 μL of $H_2O$. For tagmentation, the sample input was normalized to 0.2 ng/μL. Before the final clean-up after tagmentation and PCR, 2 μL of each sample were pooled in a single tube. The pool was cleaned-up using 0.7x SPRI ratio and sequenced in one run (40 bp paired-end mode for $Ogt^{T931A}$ and 75 bp single-end for $Ogt^{AID}$) on the Illumina NextSeq500 sequencer. S5 Table contains the number of pooled embryos and average number of reads obtained per embryo for each library.

## MEFs mRNA-Seq library preparation and sequencing

Total RNA was extracted with Monarch Total RNA Miniprep Kit (NEB #T2010S). The integrity of the RNA was determined using the TapeStation 4150 (High Sensitivity RNA kit, Agilent Technologies #5067–5579). Library preparation was performed by EMBL Genomics Core Facility using the TruSeq Stranded mRNA Library Preparation kit according to the manufacturer's instructions (Illumina #20020594). All libraries were pooled and sequenced in one run (75 bp single-end mode) on the Illumina NextSeq500 sequencer at the Genomics Core Facility of EMBL Heidelberg.

## mRNA-Seq library preparation and sequencing from single placentae

Samples were thawed and homogenized using BioSpec 11079110Z Zirconia/Silica Bead 1.0 mm Diameter (3 cycles, 90 s each, 2500 rpm), supernatant was collected in a new tube and treated 10' at 55˚C with Proteinase K (15 μL of enzyme every 300 μL of sample, provided with Monarch Total RNA Miniprep Kit). Debris were removed by centrifugation at 16000 x g for 2' and collection of the supernatant. RNA was then extracted by adding an equal volume of RNA lysis buffer and proceeding with Monarch Total RNA Miniprep Kit (NEB #T2010S) extraction, according to the manufacturer's instructions. After RNA's elution with water, an additional treatment with 5U/50μL Turbo DNAse (Thermo Scientific #AM2238) was performed for 15' at 37˚C in order to reduce DNA contamination for RNA repeats analysis. The RNA samples were cleaned with Monarch RNA Cleanup Kit (NEB #T2030) according to the manufacturer's instructions. The integrity of the RNA was determined using the TapeStation 4150 (High Sensitivity RNA kit, Agilent Technologies #5067–5579). Library preparation was performed by EMBL Genomics Core Facility using the NEBNext Ultra II Directional RNA Library Prep Kit for Illumina (NEB #E7760L), in combination with the NEBNext Poly(A) mRNA Magnetic Isolation Module for mRNA enrichment (NEB #E7490L), according to the manufacturer's

instructions. All libraries were pooled and sequenced in one run (40 bp paired-end mode) on the Illumina NextSeq500 sequencer at the Genomics Core Facility of EMBL Heidelberg.

## Cellular fractionation

About 10 million cells were collected, washed with PBS and resuspended in 10 volumes of hypotonic buffer (10 mM HEPES pH 7.65; 1.5 mM $MgCl_2$, 10 mM KCl, 0.5 mM DTT (Invitrogen #15508–013)). Cells were then incubated for 15' at 4˚C under gentle agitation and Dounce homogenized 20 times using a tight pestle. The homogenized nuclei were centrifuged at 228 x g for 5 min at 4˚C. The supernatant (cytosol) was clarified by high-speed centrifugation (5 min, 20000 x g, 4˚C). The nuclei were washed twice in 2 ml of washing buffer (15 mM HEPES pH 7.65; 10 mM $MgCl_2$, 250 mM sucrose, 0.5 mM DTT). Nuclei were lysed by resuspension in RIPA buffer (150 mM NaCl, 1% NP-40, 0.5% deoxycholic acid, 0.1% SDS, 50 mM Tris pH 8.0 in $ddH_2O$).

## Western blotting MEFs

Total protein lysates were prepared by lysing the cells with RIPA buffer (150 mM NaCl, 0.1% Triton X-100, 0.5% sodium deoxycholate, 0.1% SDS and 50nM Tris-HCl pH 8.0) and subsequent high-speed centrifugation (5 min, 20000 x g, 4˚C) to eliminate insoluble fractions. Protein concentration was quantified using the Pierce BCA Protein Quantification kit (ThermoFisher #23225). 20 µg of protein extract and 5 µL of WesternSure Pre-stained Chemiluminescent Protein Ladder (Li-cor #926–98000) were loaded in a NuPAGE 4–12% Bis-Tris or Novex 4–20% Tris-Glycine Protein Gel, 1.0 mm (ThermoFisher #NP0335BOX and #XP04200BOX) and run at 150V in an XCell Sure Lock apparatus (ThermoFisher). Proteins were transferred on a PVDF membrane in a Mini-Protean Tetra System apparatus or a Trans-Blot Turbo Transfer System (Bio-Rad). Abundant epitopes were blocked with a buffer containing 5% milk or, for O-GlcNac detection, 5% BSA (Sigma-Aldrich #A2153), and 0.1% Tween-20 in $ddH_2O$. Membranes were incubated overnight at 4˚C (or for 1 hour at room temperature for loading controls) with primary antibody dilutions in a buffer containing 5% BSA and 0.1% Tween-20 in $ddH_2O$. Membranes were washed twice with 0.5% Triton X-100 and 0.5 M NaCl in $ddH_2O$ for 5' and once with 0.5 M NaCl in $ddH_2O$ for 10', then rinsed with PBS and incubated for 1 hour at room temperature with 1:200000 secondary antibody dilutions of HRP-coupled goat anti-rabbit IgG (ThermoFisher #G-21234) or 1:100000 dilutions of HRP-coupled goat anti-mouse IgG (ThermoFisher #G-21040). Membranes were washed as after primary antibody and incubated with GE Healthcare ECL Prime (Sigma #RPN2232) for imaging using a ChemiDoc imaging system (BioRad) or ImageQuant 800 (Amersham).

The primary antibodies used were: anti-OGT (Abcam #ab177941) 1:2000, anti-O-GlcNAc clone RL2 1:2000 (Abcam #ab2739 and Merck Millipore #MABS157), anti-lamin A/C (Santa Cruz Biotechnology #sc-376248) 1:200, anti-histone H3 1:100000 (Abcam #ab1791), anti-vinculin 1:5000 (Sigma-Aldrich #V9264).

## Western blotting E12.5 placentae

Placentae were dissected at E12.5 and included three principal zones: the labyrinth, junctional zone, and decidua basalis [95]. The placental tissue was initially pestle-homogenized (ThermoFisher, #12-141-363) and then dissociated through successive trituration with gauge needles G18 and G25 in RIPA buffer with protease inhibitors (Sigma, #1136170001) and benzonase. Samples were kept under constant agitation for 2h at 4˚C followed by centrifugation at 13.300 x g at 4˚C for 20 min. The genotype was determined using the yolk sac. Protein concentration was determined using the Pierce BCA Protein Quantification kit (ThermoFisher #23225) and

25 μg of total protein extract was loaded on NuPAGE 4–12% Bis-Tris gel (ThermoFisher #NP0335BOX). Proteins were transferred onto nitrocellulose membrane by semi-dry blotting system (Trans-Blot Turbo Transfer System, Bio-Rad). The membranes were blocked in 5% BSA/TBST (Tris-buffered saline with 0.1% Tween 20) for 1 hour at room temperature before incubation with primary antibody diluted in 5% BSA/TBS-Tween overnight at 4˚C. The following day, the membranes were washed three times for 10' in TBS-Tween and incubated for 1 hour at room temperature with the appropriate HRP-coupled secondary antibodies in TBST. The lectin from *Triticum vulgaris wheat* (WGA) conjugated with the horseradish peroxidase (HRP) was used to detect O-GlcNAc-modified proteins. The membrane was incubated with WGA-HRP (100 ng/ml) for 1 hour at room temperature. Membranes were then washed three times for 10' in TBS-Tween at room temperature. PBS was used for the final wash before chemiluminescent detection of proteins with GE Healthcare ECL Prime (Sigma #RPN2232). ImageQuant 800 (Amersham) was used for imaging. Relative changes in the expression of OGT, OGA and O-GlcNAc-modified proteins were determined by measuring the ratio of the densitometric values of bands containing the above proteins (or the whole lane for O-GlcNAc) to β-Actin as a control. Fiji was used for processing the images (https://imagej.net/nih-image/manual/tech.html).

The primary antibodies used were: anti-OGT (EPR12713) (Abcam #ab177941) 0.5 μg/ml, anti-OGA (Mgea5) (ThermoFisher # PA5-67426) 1 μg/ml and anti-β-Actin antibody AC-15 (Sigma #A5441) 0.2 μl/ml. The global level of O-GlcNAc-modified proteins was determined by WGA-HRP (Sigma, #L3892-1MG). The secondary antibodies used were: HRP-coupled goat anti-rabbit IgG (ThermoFisher #G-21234) or HRP-coupled goat anti-mouse IgG (ThermoFisher, #G-21040) at concentration 1:20000.

## Cell immunofluorescence

MEFs were seeded a day before fixation in 12-well plates on coverslips covered with 0.1% gelatin. Cells were fixed for 10' at room temperature using PFA/SEM buffer (4% PFA, 0.12 M sucrose, 3 mM EGTA, 2 mM MgCl$_2$ in PBS), then washed once with PBS. Aldehyde groups were quenched with 50 mM ammonium for 10', then cells were washed once with PBS and permeabilized with 0.1% Triton X-100 for 10'. Cells were washed three times with PBS, then unspecific epitopes were saturated using 5% goat serum for 30' at room temperature. Coverslips were incubated overnight at 4˚C with primary antibody anti-OGT (Abcam #ab177941) 1:200 in 1% BSA (or with 1% BSA w/o primary antibody, as a control), the day after washed three times with PBS and incubated with the secondary antibody (A488-conjugated goat anti-rabbit IgG (ThermoFisher #A11008)) diluted 1:1000 in 1% BSA for 30' at room temperature. Coverslips were washed once with PBS, incubated with DAPI 1:1000 in PBS for 5' at room temperature and washed again with PBS. Coverslips were mounted on glass slides in ProLong Diamond Antifade Mountant (ThermoFisher #P36961). Fixed immunostained samples were imaged on a Nikon A1 confocal microscope.

## Quantification of microscopy images

All steps were performed using ImageJ [96] when not differently specified. For the quantification of O-GlcNAc signal in single morulae (S5F Fig): i. brightness and contrast of the raw images were adjusted to the same scale, ii. projection of the z stacks was performed for each image using the maximum signal intensity, iii. images were denoised using Aydin (https://doi.org/10.5281/zenodo.5654826), with Classic Image Denoiser butterworth and default parameters, iv. ellipses were drawn inside each embryo and mean intensity in the area was measured for single embryos.

## Analysis of mRNA-sequencing in MEFs and placentae for single copy genes

The analysis pipeline was performed using Galaxy [97] to obtain the table of gene counts. The quality of the reads was analyzed using FastQC v0.11.8 (https://www.bioinformatics.babraham.ac.uk/projects/fastqc/), then reads trimmed from adapters and low-quality 3'-end nucleotides using Trim Galore v0.6.3 (https://www.bioinformatics.babraham.ac.uk/projects/trim_galore/) with default parameters for single-end or paired-end libraries (-q 20—stringency 1 -e 0.1—length 20 –paired). Reads were mapped to Gencode vM25 (GRCm38.p6) transcript sequences using Salmon v0.8.2 [98] with default parameters for stranded single-end or for paired-end libraries. For MEFs, the *AID-2xMyc-FLAG* tag and the *OsTIR-Myc-HA* transgene had been added to the transcriptome. The gene counts were used in a custom Rmd script as input for DESeq2 v1.34.0 [99] after gene-level summarization using tximport [100]. The test used for statistical significance was the Wald test, the obtained p-values were corrected for multiple testing using the Benjamini and Hochberg method (default in DESeq2) and the significance cutoff for optimizing the independent filtering was 0.05. Whenever MA-plots are shown, the $\log_2$ fold changes are shrunken using the 'ashr' method [101] and the x-axis shows mean DESeq2-normalized gene counts across samples.

## Analysis of single embryo mRNA-sequencing for single copy genes

The analysis pipeline was performed using Galaxy [97] to obtain the table of gene counts. Briefly, the quality of reads was analyzed using FastQC v0.11.8 (https://www.bioinformatics.babraham.ac.uk/projects/fastqc/). Reads were trimmed from adapters and low-quality 3'-end nucleotides using Trim Galore v0.6.3 (https://www.bioinformatics.babraham.ac.uk/projects/trim_galore/) with default parameters for single-end or paired-end libraries (-q 20—stringency 1 -e 0.1—length 20 –paired), before mapping them to the GRCm38 mouse genome using STAR v2.7.8a [102] and default parameters for single-end or paired-end reads. For the *Ogt^AID* blastocysts, the *AID-2xMyc-FLAG* tag and the *OsTIR-Myc-HA* transgene had been concatenated to the genome fasta and GTF annotation. Gene counts were obtained from the bam files using featureCounts (subreads v2.0.1) [103] with default parameters for single-end or paired-end reads and counting fragments instead of reads in the latter case. The gene counts were used in a custom Rmd script as input for DESeq2 v1.34.0 [99]. The test used for statistical significance was the Wald test, the obtained p-values were corrected for multiple testing using the Benjamini and Hochberg method (default in DESeq2) and the significance cutoff for optimizing the independent filtering was 0.05.

For the *Ogt^AID* blastocysts, pooled from more than one embryo generation and collection experiment, batch effect had to be considered when testing for differential expression using 'DESeq' function. To this aim, the package RUVSeq [104] was used to compute the factors of unwanted variation, with function 'RUVg' and k = 6. The subset of control genes to use in RUVg was found with a first run of DESeq2 differential expression analysis, separately for all untreated and all auxin-treated blastocysts: the genes with i. mean DESeq2-normalized counts across samples > 100, ii. adjusted p-value > 0.8 and iii. $\log_2$FC < 0.05 for each condition were selected. The 6 factors of unwanted variation computed with RUVSeq were used in 'DESeq' formula ~ *W1* +. . . + *W6* + *genotype*.

For MA-plots, the $\log_2$ fold changes are shrunken using the 'ashr' method [101] and the x-axis shows mean DESeq2-normalized gene counts across samples. Before principal component analysis (PCA) and differential expression analysis, low-quality samples were removed by discarding embryos with < $10^6$ reads and outlier embryos in a scatter plot of mitochondrial DNA (mtDNA) gene expression versus percentage of reads mapping to ribosomal DNA (rDNA). The exact number of embryos filtered at each step and the final number used for DE analysis is in S4 Table.

### *In silico* genotyping of single embryos from mRNA-sequencing data

Sex was assigned to the single embryo transcriptomes based on the DESeq2-normalized counts of chrY-mapping genes *Ddx3y* and *Eif2s3y* [105]. The assignment of the *Ogt*$^{AID}$ genotype was based on the raw sum of reads mapping to the *AID-2xMyc-FLAG* exogenous sequence, whose distribution divided the samples in two distinct populations. A sample was defined as wild type if the sum of *AID-2xMyc-FLAG*-mapping reads was $< 72$.

### Analyses of gene set enrichment from mRNA-sequencing data

All performed in a custom Rmd script. Gene ontology (GO) over-representation test in MEFs was performed on differentially expressed genes (DEGs) (adj. p-value $< 0.05$, any log$_2$FC) with mean DESeq2-normalized counts across samples $> 10$ using function 'enrichGO' of R package clusterProfiler v4.2.2 [106] with adjusted p-value and q-value cutoffs of 0.05 and 0.1, respectively, and default parameters. GO level 5 was then selected using function 'gofilter' and results were simplified by q-value using function 'simplify' and a similarity cutoff of 0.6.

for MEFs) or 0.7 (for blastocysts from the *Ogt*$^{T931A}$ allele's IVF experiment).

Analysis of over-representation of Molecular Signature Database (MsigDb) hallmark gene sets was performed on DEGs with mean DESeq2-normalized counts across samples $> 10$ using function 'GSEA' of R package clusterProfiler v4.2.2 with adjusted p-value cutoff of 0.05 and default parameters.

Gene set enrichment analysis (GSEA) was performed on all genes of the dataset with mean DESeq2-normalized counts across samples $> 10$, ranked by -log10(adj. p-value)*sign(log$_2$FC), using function 'gseGO' (for GO terms) or GSEA (for placental markers and for Fig 2F) of R package 'clusterProfiler' v4.2.2, with adjusted p-value cutoff of 0.05 and default parameters. For GO terms, GSEA results were simplified based on adj. p-value after computing semantic similarity using 'mgoSim' function of R package 'GoSemSim' [107]. Similarity cutoff was 0.6 for all figures showing GSEA results with GO terms.

### Principal component analysis of single embryo mRNA-sequencing data

The PCAs shown in Figs 2C, S2B, S2C and S5H were produced with function 'prcomp' on log$_2$-transformed, DESeq2-normalized counts, using the first 1000 genes with the highest variance, after removing genes with mean DESeq2-normalized counts across samples $\leq 10$.

### Analysis of retrotransposons' expression

The analysis was performed using a custom Snakemake v5.9.1 pipeline [108], available at https://github.com/boulardlab/Ogt_mouse_models_Formichetti2024.

In summary, the quality of the fastq files was checked with FastQC v0.11.8 and reads were trimmed using Trim Galore v0.6.4 with default parameters for paired-end libraries (-q 20—stringency 1 -e 0.1—length 20 –paired). Trimmed reads were aligned to GRCm38 mouse genome with STAR v2.7.5c [102], with parameters recommended by Teissandier *et al.* [109] for the analysis of transcripts derived RNA TEs in the mouse genome:—outFilterMultimapN-max 5000—outSAMmultNmax 1—outFilterMismatchNmax 3—outMultimapperOrder Random—winAnchorMultimapNmax 5000—alignEndsType EndToEnd—alignIntronMax 1—alignMatesGapMax 350—seedSearchStartLmax 30—alignTranscriptsPerReadNmax 30000—alignWindowsPerReadNmax 30000—alignTranscriptsPerWindowNmax 300—seedPerReadNmax 3000—seedPerWindowNmax 300—seedNoneLociPerWindow 1000. After mapping, using a custom script, reads were kept only from pairs where both mates were:—completely included into a repetitive element;—not overlapping with gene bodies of Gencode

vM25. Repeat Library 20140131 (mm10, Dec 2011) was used as repetitive elements annotation, after excluding "Simple repeats" and "Low complexity repeats".

With the STAR parameters above, only one random alignment (that with the highest alignment score) is reported for multimappers, preventing the precise quantification of single repetitive elements. Therefore, for each repName of the repetitive element annotation—often present in multiple copies in the genome—read counts were summarized using featureCounts (subreads v2.0.1) [103], with parameters -p -B -s 0—fracOverlap 1 -M.

The downstream analysis of TEs expression was performed in a custom Rmd script. First, for Figs 2G and S2F, an analysis at the family level (repFamily field in the Repeat Library annotation) was performed. For this analysis, Fragments per Kilobase per Million (FPKM) were computed for each repFamily by: i. summing counts for all elements belonging to that repFamily; ii. dividing this sum by the number of STAR input reads for that sample (/2 because of featureCounts quantification of fragments instead of single reads) and multiplying by $10^6$; iii. dividing what obtained by the total sum of featureCounts-summarized lengths (in Kb) of all elements belonging to that repFamily. FPKM values of DNA transposon families were manually checked to remove samples with high DNA contamination. Differential Expression analysis of RNA TEs was performed at the repName level, using DESeq2 v1.34.0 [99] and adding the value of the total sum of DNA FPKM as a confounding factor to DESeq formula (~ *DNA_FPKM + condition*). The test used for statistical significance was the Wald test, the obtained p-values were corrected for multiple testing using the Benjamini and Hochberg method (default in DESeq2) and the significance cutoff for optimizing the independent filtering was 0.05. For MA-plots, the $\log_2$ fold changes are shrunken using the 'ashr' method [101] and the x-axis shows mean DESeq2-normalized counts. For S2G and S3D Figs, FPKM values for each repName were computed as follows: i. repName counts (from featureCounts summarization) were divided by the total number of STAR input reads for that sample (/2 because of featureCounts quantification of fragments instead of single reads) and multiplied by $10^6$; ii. what obtained was divided by the featureCounts-summarized length (in Kb) of that repName.

### Analysis of publicly available mRNA-Seq data

The analysis of datasets GSE66582 [110] and GSE76505 [58] for Figs 2F, S2E and S2H was performed as described in [50].

### Statistical analyses

All statistical tests used are specified in the figure legends. The exact p-values are indicated in the figures, while adjusted p-values are indicated in figures only when <0.05 for single copy genes or placental retrotransposons or <0.1 for preimplantation embryos's retrotransposons and higher values are considered not significant. For boxplots, hinges correspond to first and third quartiles; median is shown inside; whiskers extend to the largest and smallest values no further than 1.5 * IQR from the hinge (IQR = inter-quartile range, or distance between the first and third quartiles); data beyond the end of the whiskers are plotted individually.

### Supporting information

**S1 Fig. Mutation of the putative NLS has no detectable effect on OGT nuclear localization.** (**A**) Linear representation of the domain composition of OGTp110 (UniProt Q8CGY8), indicating in red the position of the putative nuclear localization signal $DFP_{461-463}$ [49]. The mouse allele $Ogt^{NLS-}$ bears an 8 pb substitution, resulting in the $DFP_{461-463}$->$AAA_{461-463}$ amino acid substitution. TPR: N-terminal tetratricopeptide repeat; N/C-Catalytic: N- and C-terminal lobes of the catalytic domain. (**B**) Representative images of OGT

immunofluorescence staining (ab177941 antibody) in WT and $Ogt^{NLS-/Y}$ male primary MEFs, showing nuclear signal also in cells where the putative NLS is mutated. The result was confirmed using two pairs of WT and $Ogt^{NLS-/Y}$ MEF clones. One z plane is shown. Scale bar indicates 20 μm. (**C**) Western blot of endogenous OGT protein (ab177941 antibody) in $Ogt^{NLS-/Y}$ and WT male primary MEFs after cellular fractionation. Vinculin (VCL) and histone H3 were probed as controls for the cytosolic and nuclear fractions, respectively. The amount of $OGT^{NLS-}$ in the nuclear compartment is comparable to WT OGT. The result was reproduced using a different pair of WT and $Ogt^{NLS-/Y}$ MEF clones.
(EPS)

**S2 Fig. Deregulation of genes and retrotransposons in blastocysts with mutations of *Ogt* at T931.** (**A**) Integrative Genomics Viewer (IGV) screenshot showing RNA-Seq reads around the T931 residue of *Ogt* for three representative genotypes of male blastocysts found in the dataset. Mutated nucleotides are colored in their containing reads or black barred when deleted. For the T931del mutant, reads marked with an asterisk were soft-clipped by STAR, which indicates that the last three mapped nucleotides (CAC) are, in fact, coming from the next three ones (also CAC) of the cDNA, due to the deletion. Note that amino acids 931 and 932 are both Threonines (T). (**B**) Transcriptomes of individual male blastocysts produced by the experiment described in Fig 2B in the space defined by PC1 and PC2 of their PCA. The variance explained by each PC is in parentheses. (**C**) PCA of transcriptomes of individual female blastocysts produced by the experiment described in Fig 2B. (**D**) DESeq2-normalized counts of *Ogt* for all blastocysts produced in the experiment in Fig 2B. P-values are from DESeq2 Wald test. (**E**) Expression dynamics of the 10 most significant (based on DESeq2 p-value) upregulated and downregulated $Ogt^{T931A/Y}$ DEGs throughout preimplantation development (mRNA-Seq data from GSE66582 and GSE76505). The two biological replicates per stage were averaged. For each gene, the TPM value in the E3.5 blastocyst is the highest TPM value between the values in E3.5 ICM and E3.5 trophectoderm. The mean among all genes is drawn, as well as the 95% confidence interval, computed using basic nonparametric bootstrap (R function 'mean.cl. boot'). Y-axis ticks are in $\log_2$ scale. TPM: Transcript Per Million. (**F**) PCA biplot of the FPKM expression values of the main families of retrotransposons, in male blastocysts from the experiment described in Fig 2B. The variance explained by each PC is in parentheses and repeat families are coloured by their repeat class. N = 9 WT, 10 $Ogt^{T931A/Y}$ embryos. (**G**) Expression (FPKM) of the most significant (based on DESeq2 p-value) differentially expressed (adj. p-value $< 0.1$, any $\log_2$FC, mean DESeq2-normalized gene counts $> 10$) retrotransposons in $Ogt^{T931A/Y}$ versus WT male blastocysts, in all blastocysts produced in the experiment described in Fig 2B. (**H**) Scatter plot comparing the transcriptional deregulation of retrotransposons in $Ogt^{T931A/Y}$ versus WT male blastocysts with their expression dynamics between the 8-cell and E3.5 blastocyst stage in unperturbed embryos. The log2FC at E3.5 vs. 8-cell stage was computed as (75% log2FC E3.5 trophectoderm vs. 8-cell) + (25% log2FC E3.5 ICM vs. 8-cell). Repeats are colored by family and labeled only if significant (adj. p-value $< 0.1$) in the $Ogt^{T931A/Y}$ vs. WT comparison. Pearson correlation coefficient is shown both for significantly deregulated repeats and for all repeats in the scatter plot.
(EPS)

**S3 Fig. Developmental phenotype and deregulation of retrotransposons upon a mild reduction of OGT's activity.** (**A**) MA-plot from DESeq2 differential expression analysis of single copy genes in all female versus all male placentae in the dataset (all genotypes; N = 11 female, 12 male placentae). All genes with adj. p-value $< 0.05$, any $\log_2$FC are colored based on their location on sex or autosomal chromosomes, and their number is indicated. Genes standing out (and with abs($\log_2$FC) $\geq 0.2$) are labeled. Genes previously reported as sexually

differentially expressed in E12.5-E18.5 placentae are in bold [33]. (**B**) Average DESeq2-normalized counts of marker genes for LaTP and SynTII labyrinth clusters in single placentae with all four genotypes analyzed. (**C**) (left) Representative images of E11.75 and E12 WT embryos among the ones used for developmental staging based on forelimb shape using the eMOSS software (https://limbstaging.embl.es/) and (right) result of the staging for all litters analyzed. N = 6 embryos dissected at E11.75 and 8 embryos dissected at E12.5 from a total of two litters from WT mothers, 16 embryos dissected at E12.5 from two litters from $Ogt^{Y851A/+}$ mothers. (**D**) MA-plots from DESeq2 differential expression analysis of retrotransposons in (left) $Ogt^{Y851A}$-homozygous versus heterozygous female placentae and (right) $Ogt^{Y851A}$-hemizygous versus WT male ones. All repeats with mean DESeq2-normalized gene counts > 10, adj. p-value < 0.05, any $\log_2$FC are colored and labeled. (**E**) Expression (FPKM), in all four placental genotypes analyzed, of the four significantly upregulated (adj. p-value < 0.05, any $\log_2$FC) retrotransposons in $Ogt^{Y851A/Y}$ versus WT male placentae together with two of the most active IAP elements (IAP-d-int and IAPEz-int, normally repressed and expressed in murine trophoblast stem cells, respectively [112]), which are not significantly upregulated. The mean for each placental genotype is drawn. padj = adj. p-value computed using DESeq2 Wald test and corrected for multiple testing using the Benjamini and Hochberg method. (**F**) Sum of FPKM of the only two types of satellite repeats (Repeat Library 20140131 for mm10, Dec 2011)) that are detected in the RNA-Seq dataset. They are both upregulated in male $Ogt^{Y851A/Y}$ placentae. The mean for each placental genotype is drawn. P-value was computed using the unpaired Wilcoxon rank sum exact test on the FPKM after subtracting the sum of FPKM of all DNA repetitive elements, in order to account for DNA contamination. In (**D**-**F**), N per genotype = 6 placentae for female homozygous and male WT, 5 for female heterozygous (due to exclusion of one outlier sample after unsupervised clustering) and male homozygous (due to exclusion of one sample with higher counts from DNA repetitive elements), all genotypes coming from at least two different litters.
(EPS)

**S4 Fig. Rapid degradation of endogenous OGT in MEFs using the AID system.** (**A**) Western blot detection of endogenous OGT in whole cell protein extracts from female and male E7.5 embryos WT or bearing the AID-*Ogt* allele. (**B**) Scheme of the mouse cross used to produce double mutant E12.5 embryos (*OsTIR*,$Ogt^{AID}$) and littermate controls (*OsTIR*,$Ogt^{WT}$) for MEFs derivation and *in vitro* auxin-induced AID-OGT degradation. (**C**) Quantification of the OGT western blot in Fig 4C. The intensity of the OGT band was normalized to LAMIN C intensity. Bar plot heights and error bars show the mean and standard deviation, respectively, of two biological replicates using two pairs of *OsTIR*,$Ogt^{AID}$, *OsTIR*,$Ogt^{WT}$ littermate clones from different litters. (**D**) Western blot of endogenous OGT (ab177941 antibody) in whole cell protein extracts from male primary MEFs, untreated or treated with auxin for the same amount of time as for RNA-Seq analysis. Lamin A/C was probed as a loading control. (**E**) Western blot analysis of the kinetic of O-GlcNAc depletion following OGT acute degradation in MEFs, using the RL2 anti-O-GlcNAc antibody.
(EPS)

**S5 Fig. Inefficient O-GlcNAc perturbation using the AID-OGT degron system in the preimplantation embryo.** (**A**) Scheme of the experiments performed to test AID-OGT degradation *ex vivo* in preimplantation embryos. Top: IVF between WT sperm and either $Ogt^{WT/WT}$ or $Ogt^{AID/WT}$ *OsTIR*-expressing females was used to produce embryos untreated or treated with auxin from the fertilization plate, which were stained for OGT and O-GlcNAc at the zygote and morula stages, respectively. Bottom: natural mating of $Ogt^{AID/WT}$ females with *OsTIR*-homozygous males was used to produce zygotes grown *ex-vivo* until the blastocyst

stage for single embryo mRNA-Seq; 24 hours before collection, half of them were moved to a medium supplemented with 250 μM auxin. In both types of experiment, females heterozygous and males homozygous AID-*Ogt* as well as control *Ogt* WT embryos, all expressing *OsTIR*, are produced for analysis. (**B**) Test for auxin toxicity on WT embryos. Embryos were generated through IVF and cultured *ex vivo* in the absence or presence of auxin from the moment of fertilization to E4. Representative widefield microscopy images are shown for both conditions, total number of starting zygotes and percentage of E4 expanded blastocysts are indicated. Scale bar indicates 40 μm. (**C**) DESeq2-normalized gene counts of *OsTIR1* for E4 blastocysts obtained from IVF of *OsTIR*,*Ogt*$^{AID}$ (N = 19 blastocysts) or control *Ogt*$^{AID}$ oocytes (not bearing the *OsTIR1* gene; N = 15 blastocysts) with WT sperm. The mean for each group of embryos is drawn. (**D**) Percentage of E4 blastocysts obtained from the same IVF as in (**C**). Embryos of both groups were treated with auxin from the time of fertilization to E4. Barplot heights and error bars indicate the mean and standard deviation, respectively, of three replicate experiments of IVF followed by auxin treatment. Total number of starting 2-cell embryos for the two groups is stated in the legend. P-value is for paired two-sided Student's t-test, assuming unequal variance. (**E**) Representative images of O-GlcNAc immunofluorescence staining (RL2 antibody) in morulae from *OsTIR*,*Ogt*$^{AID/WT}$ and control *OsTIR*,*Ogt*$^{WT/WT}$ oocytes, grown in 250 μM auxin from the time of fertilization. Embryos were mounted in drops and imaged using an X-light V3 Spinning disk confocal. One z plane is shown. Scale bar indicates 20 μm. The total number of imaged embryos is indicated. (**F**) Quantification of the O-GlcNAc fluorescence signal from the morulae in (**A**) (Methods). Average signal per group is marked. P-value is for unpaired two-sided Student's t-test, assuming unequal variance. (**G**) Representative images of OGT immunofluorescence staining (ab177941 antibody) in zygotes from IVF of *OsTIR*,*Ogt*$^{AID/WT}$ oocytes with WT sperm, untreated or treated with 500 μM auxin from the time of fertilization. Embryos were mounted in slides and imaged using an X-light V3 Spinning disk confocal. One z plane is shown. Scale bar indicates 20 μm. The total number of imaged embryos is indicated. (**H**) PCA of E4.5 blastocysts from the RNA-Seq experiment depicted in (**A**). The variance explained by each PC is in parentheses. (**I**) For the same RNA-Seq experiment as in (**H**), MA-plots from DESeq2 differential gene expression analysis of *Ogt*$^{AID}$ versus *Ogt*$^{WT}$ genotypes' comparison, separately for male and female blastocysts, untreated and treated with auxin. Genes with mean DESeq2-normalized gene counts > 10, adj. p-value < 0.05, any log$_2$FC (DEGs) are colored, and their number is indicated. Genes with abs(log2FC) ≥ 0.2 are labeled. Among the 32 upregulated DEGs, 4 were also significantly upregulated at a higher degree in *Ogt*$^{T931del/Y}$ blastocysts: *Washc3*, which promotes actin polymerization at the surface of endosomes, the lysosomal transporter *Mfsd1*, mitochondrial insertase *Mtch2* and isocitrate dehydrogenase 1 (*Idh1*). In (**H-I**), N = 13 and 16 treated and untreated females *Ogt*$^{AID}$, 14 and 12 treated and untreated females *Ogt*$^{WT}$, 9 and 13 treated and untreated males *Ogt*$^{AID}$, 11 and 11 treated and untreated males *Ogt*$^{WT}$.
(TIF)

**S6 Fig. Kinetics of differential gene expression after rapid degradation of endogenous OGT in MEFs.** (**A**) Volcano plots from DESeq2 analysis of gene expression changes in (left) untreated *OsTIR*,*Ogt*$^{AID}$ versus untreated *OsTIR*,*Ogt*$^{WT}$ MEFs clones (effect of the hypomorphic genotype) and (right) *OsTIR*,*Ogt*$^{WT}$ control clones treated with auxin for 96 hours versus grown untreated (effect of the auxin drug). Differentially expressed genes with p-value < 10$^{-5}$ and absolute log$_2$FC > 0.3 (i.e. 1.2-fold increase or decrease in expression) are labeled in red and their number is indicated. (**B**) DESeq2-normalized counts of *Ogt*, *Oga* and the transgene *OsTIR1*. *Oga* level was already downregulated in the untreated *OsTIR*,*Ogt*$^{AID}$ genotype, which is *de facto* an hypomorphic *Ogt* mutant, when compared with the *OsTIR*,

$Ogt^{WT}$ one. Note that $OsTIR1$ is also significantly downregulated after auxin addition, suggesting that an unknown mechanism counteracts the degron system upon auxin treatment. Mean of counts for each group of samples is shown. Y-axis ticks are in $\log_{10}$ scale. padj = adj. p-value computed using DESeq2 Wald test and corrected for multiple testing using the Benjamini and Hochberg method. (**C**) Number and overlap of differentially expressed genes (DEGs; adj. p-value < 0.05, any $\log_2$FC) in auxin-treated $OsTIR,Ogt^{AID}$ clones versus untreated $OsTIR,Ogt^{AID}$ clones at the three time points analyzed. DEGs in auxin-treated control clones at any time point are excluded. (**D**) Gene ontology (GO) over-representation analysis of DEGs (adj. p-value < 0.05, any $\log_2$FC) for the same comparisons between genotypes and conditions as in Fig 4E, but showing result for Cellular Component (CC) and Molecular Function (MF) GO terms. The first 25 most enriched CC and MF GO terms are shown, based on p-value across all comparisons. Terms are ordered by gene ratio. Gene ratio = genes belonging to the GO term / total number of deregulated DEGs for that comparison. UP = upregulated DEGs, DOWN = downregulated DEGs. Terms enriched due to auxin treatment on control clones are written in gray.
(EPS)

**S1 Table. Description of the murine alleles generated.**
(DOCX)

**S2 Table. List of primers for genotyping the murine alleles.**
(DOCX)

**S3 Table. List of primers for sexing and genotyping the MEFs.**
(DOCX)

**S4 Table. List of primers for sexing and genotyping Ogt$^{T931A}$ blastocysts.**
(DOCX)

**S5 Table. Details on the generation and filtering steps of the single embryo Smart-Seq datasets.**
(DOCX)

**S1 Text. The AID-OGT degron system is inefficient in ex vivo grown preimplantation embryos.**
(DOCX)

## Acknowledgments

We thank members of the Boulard laboratory for helpful discussion and Agnese Loda for critical reading of the manuscript. We thank the EMBL Rome Laboratory Resources Animal Facility and in particular Giuseppe Chiapparelli and Valerio Rossi for their essential support with animal husbandry and genotyping. We acknowledge all members of the Gene Editing and Virus Facility at EMBL Rome for assistance with mouse genome engineering, the EMBL Rome microscopy facility for assistance with microscopy imaging and the EMBL Genomic Core Facility for the preparation of the mRNA-Seq libraries and next generation sequencing. We are grateful to Masato Kanemaki for kindly providing us with the plasmid containing the $OsTIR1$ cDNA. We also thank Javier Lizarrondo for the cartoon representation of OGT's structure and Francesco Tabaro for invaluable support with code sharing.

## Author Contributions

**Conceptualization:** Sara Formichetti, Mathieu Boulard.

**Data curation:** Sara Formichetti.

**Formal analysis:** Sara Formichetti, Julia Hansen.

**Funding acquisition:** Mathieu Boulard.

**Investigation:** Sara Formichetti, Agnieszka Sadowska, Michela Ascolani, Julia Hansen.

**Methodology:** Sara Formichetti, Agnieszka Sadowska, Michela Ascolani, Julia Hansen, Neil Humphreys, Mathieu Boulard.

**Project administration:** Sara Formichetti, Mathieu Boulard.

**Resources:** Kerstin Ganter, Christophe Lancrin, Mathieu Boulard.

**Software:** Sara Formichetti.

**Supervision:** Christophe Lancrin, Neil Humphreys, Mathieu Boulard.

**Validation:** Sara Formichetti, Agnieszka Sadowska, Michela Ascolani, Neil Humphreys.

**Visualization:** Sara Formichetti, Mathieu Boulard.

**Writing – original draft:** Sara Formichetti, Michela Ascolani, Neil Humphreys, Mathieu Boulard.

**Writing – review & editing:** Sara Formichetti, Mathieu Boulard.

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
