## [Decision Letter · Decision Letter 0]

30 Sep 2024

Dear Dr Boulard,

Thank you very much for submitting your Research Article entitled 'Genetic gradual reduction of OGT activity unveils the essential role of O-GlcNAc in the mouse embryo' to PLOS Genetics.

The manuscript was fully evaluated at the editorial level and by independent peer reviewers. The reviewers appreciated the attention to an important topic but identified some minor areas that we ask you address in a revised manuscript.

We therefore ask you to modify the manuscript according to the review recommendations. Your revisions should address the specific points made by each reviewer.

To resubmit, log into your Editorial Manager account and select the option 'Revise Submission' in the 'Submissions Needing Revision' folder.

Yours sincerely,

Marnie E. Blewitt

Academic Editor

PLOS Genetics

John Greally

Section Editor

PLOS Genetics

Reviewer's Responses to Questions

**Comments to the Authors:**

Reviewer #1: In this manuscript, the authors demonstrated that catalytic activity of OGT is crucial for embryonic development by using mice having single amino acid mutations on its catalytic domain. The mutant mice showed altered gene expression including upregulation of some retrotransposons, which may explain developmental delay and sub lethality of the mutant embryos.

The experiments are conducted well, but the manuscript is long and often complicated. I would recommend that the authors edit the manuscript in more concise style. Descriptions are often confusing as listed below, which should be edited. In vivo degron experiments are not informative and should be deleted from this manuscript.

Specific comments:

1. Page 3, 1st paragraph; ‘The lack of maternal transmission of the OgtQ849N allele from seemingly mosaic founder females is likely explained by the reliance of the cleavage stage embryo on the oocyte payload of OGT and O-GlcNAc modified proteins.’

It is not clear for me what this sentence means, because mosaic founders seem to give rise to wild type embryos as well.

2. Page 6, 2nd paragraph; ‘especially considering that a posteriori sequencing-based genotyping of the pair of control mothers revealed that only one of them was OgtT931del/+ while the other WT.’

It is not clear for me what this sentence means. Why control (wild type?) mothers give rise to OgtT931del/+ ?

3. Page 8, 3rd paragraph, Fig. 3D; ‘the total O-GlcNAc level was reduced in hemizygous males and homozygous females (Fig. 3C,D)’

Is it statistically significant?

4. Page 9, 1st paragraph; ‘Markers for precursor cell types of both labyrinth and junctional zone (LaTPs and JZPs) were all upregulated in OgtY851A/Y placentae, while markers for matured cell types (SynTs, sTGCs, SpT), which increase in percentage from E10.5 to E12.5 [69], were all downregulated (Fig. 3I). Of note, downregulation was also observed for markers of endothelial cells, which also compose the fetal-maternal exchange barrier but are not of trophoblast origins (Fig. 3I).’

This sentence is unclear in terms of correlation to Fig.3I. Referring Fig.3H may be more suitable.

5. Page 10-11, Fig.S5; Because AID-OGT experiments in vivo are not informative due to inefficient AID degron, I recommend to remove this part.

6. Page 13, 5th paragraph; ‘The developmental requirement of OGlcNAcylation was not obvious’

It is not clear for me what this sentence means.

Reviewer #2: The manuscript by Formichetti et al. addresses a pertinent issue regarding the developmental role of O-GlcNAc in mouse development. Building upon their prior experiments, the authors have performed an impressive amount of work to establish mouse models that allow for the dissection of the catalytic function of OGT rather than the whole complex. Importantly, the manuscript reports on mouse mutants showing a range of catalytic activities.

Using these new tools, the authors first focus on the early developmental phenotypes of the severe hypomorph mutant. This analysis revealed that fewer blastocysts carrying the mutation are found in both males and females, which relates to significant dysregulation of several genes, including those linked to metabolism. Similarly, there is mild transposable element upregulation. Overall, mutant embryos show signs of developmental delay and metabolic dysregulation. In the case of a mild reduction in OGT activity, the authors report a reduced litter size and sex-specific placental phenotypes. Specifically, male placentas show significant dysregulation of multiple pathways and signs of delayed placental differentiation.

Overall, this is an incredible amount of mouse work that reports on interesting and novel OGT-associated phenotypes. I only have minor points of criticism that could be addressed before publication:

1. Figure 2F/S2E reports some developmental delay; however, the analysis seems a bit cherry-picked. I would suggest performing GSEA where all genes are ranked based on their developmental dynamics between 8c and E3.5. Next, one can look at differentially expressed genes in mutants and see if they are enriched for such developmental genes.

2. In general, the TE upregulation is mild, and there are also some downregulated TEs. Can we really say that there is mild upregulation? It’s rather mild mis-regulation.

3. One limitation of the study is that the levels of O-GlcNAc are not always quantified. For T931A, there is no quantification (even by IF), while for Y851A, there is a nice WB, but I don’t understand the quantification (Fig 3D). Why is the effect so mild in the quantification? The authors also state: “the lower sensitivity of homozygous females’ transcriptome to Ogt disruption seems difficult to reconcile with their comparable (lower) O-GlcNAc levels to hemizygous males.” However, I don’t see these lower levels. What is this relating to?

4. Authors mention that T931del shows signs of translational stress, but I don’t understand what this is based on.

5. Some explanation of what Fig 3G represents would be helpful.

Stylistically, I would like to see this text shortened, and I also miss some conclusions after individual parts. Nevertheless, this is an interesting study that seems suitable for PLOS Genetics.

Reviewer #3: Formichetti and colleagues put forward an interesting, thorough investigation of the importance of OGT in vivo. This is an effort marked by many challenges, some of which unsurmountable within the time frame and the current state of affairs of competitive fundamental research involving mouse models.

Due to OGT being essential for development, the author's efforts to characterize a series of alleles, some of which allowing development, have allowed unprecedented advances in characterizing the consequences of OGT defects in mutants. Incidentally, together with the difficulties of working in vivo, this makes the interpretation of data much more tricky as the effects are milder and the complexity is high. The authors however stuck to high level scientific rigor and did an amazing job making claims that are true to their data, being open about the shortcomings, and discussing the possibilities of what the data means in light of what is known.

This work has also been seen and reviewed by other 4 reviewers, whose questions that where within reasonable scope were addressed and discussed carefully by the authors.

In light of all this, the opinion of this 5th reviewer is that this work has to be accepted and published. It is a large effort for a fairly small conceptual advance, but it is a real advance and it does set a new high standard for future efforts aiming at characterizing the role of OGT in mammalian development.

I would like to congratulate the authors on the great work and sticking to a good scientific reporting of difficult findings.

**Have all data underlying the figures and results presented in the manuscript been provided?**

Reviewer #1: Yes

Reviewer #2: Yes

Reviewer #3: Yes

PLOS authors have the option to publish the peer review history of their article (what does this mean?). If published, this will include your full peer review and any attached files.

Reviewer #1: No

Reviewer #2: No

Reviewer #3: **Yes: **Joan Barau

---

## [Editor Report · Decision Letter 1]

18 Nov 2024

Dear Dr Boulard,

We are pleased to inform you that your manuscript entitled "Genetic gradual reduction of OGT activity unveils the essential role of O-GlcNAc in the mouse embryo" has been editorially accepted for publication in PLOS Genetics. Congratulations!

Yours sincerely,

Marnie E. Blewitt

Academic Editor

PLOS Genetics

John Greally

Section Editor

PLOS Genetics

Aimée Dudley

Editor-in-Chief

PLOS Genetics

Anne Goriely

Editor-in-Chief

PLOS Genetics

Comments from the reviewers (if applicable):

**Data Deposition**

http://datadryad.org/submit?journalID=pgenetics&manu=PGENETICS-D-24-00917R1

**Press Queries**

---

## [Editor Report · Acceptance letter]

22 Nov 2024

PGENETICS-D-24-00917R1 

Genetic gradual reduction of OGT activity unveils the essential role of O-GlcNAc in the mouse embryo 

Dear Dr Boulard, 

We are pleased to inform you that your manuscript entitled "Genetic gradual reduction of OGT activity unveils the essential role of O-GlcNAc in the mouse embryo" has been formally accepted for publication in PLOS Genetics! Your manuscript is now with our production department and you will be notified of the publication date in due course.

With kind regards,

Anita Estes

PLOS Genetics

On behalf of:
